# New Proglacial Meteorology and River Stage Observations from Inglefield Land and Thule, NW Greenland

Sarah E. Esenther[1, 2], Laurence C. Smith[1, 2], Adam LeWinter[3], Lincoln H Pitcher[4,5], Brandon T. Overstreet[6], Aaron Kehl[3], Cuyler Onclin[7], Seth Goldstein[2], Jonathan C. Ryan[8]

[1]Department of Earth, Environmental, and Planetary Sciences (DEEPS), Brown University, Providence, Rhode Island, 02912, USA

[2]Institute at Brown for Environment and Society (IBES), Brown University, Providence, Rhode Island, 02912, USA

[3]U.S. Army Corps of Engineers, Cold Regions Research and Engineering Laboratory, Hanover, New Hampshire, 03755, USA

[4]Cooperative Institute for Research in Environmental Sciences (CIRES), University of Colorado (CU) Boulder, Boulder, CO, 80309, USA

[5]Oak Ridge Institute for Science and Education (ORISE), Oak Ridge, TN, 37830, USA

[6]Department of Geology and Geophysics, University of Wyoming, 1000 E. University Ave. Laramie, WY, 82071, USA

[7]Consultant, Hydrology Technologist and Field Safety specialist, Saskatoon, Saskatchewan, S7J2S1, Canada

[8]Department of Geography, University of Oregon, Eugene, Oregon 97401, USA

*Correspondence to*: Sarah E. Esenther (sarah_esenther@brown.edu)

**Abstract.** Meltwater runoff from the Greenland Ice Sheet (GrIS) is an important contributor to global sea level rise, but substantial uncertainty exists in its measurement and prediction. Common approaches for estimating ice sheet runoff are in situ gauging of proglacial rivers draining the ice sheet, and surface mass balance (SMB) modeling. To obtain hydrological and meteorological datasets suitable for both runoff stage characterization and, pending establishment of stage-discharge curves, SMB model evaluation, we established an automated weather station (AWS) and cluster of traditional and experimental river stage sensors on the Minturn River, the largest proglacial river draining Inglefield Land, NW Greenland. Secondary installations measuring river stage were installed in the Fox Canyon River and North River at Thule Air Base, NW Greenland. Proglacial runoff at these sites is dominated by supraglacial processes only, uniquely advantaging them for SMB studies. The three installations provide rare hydrological time-series and an opportunity to evaluate experimental measurements of river stage from a harsh, little-studied polar region. The installed instruments include submerged vented and non-vented pressure transducers, a bubbler sensor, experimental bank-mounted laser rangefinders, and time-lapse cameras. The first three years of observations (2019 to 2021) from these stations indicate a) a meltwater runoff season from late June to late August/early September, roughly synchronous throughout the region; b) early onset (~June 23 to July 8) of a strong diurnal runoff signal in 2019 and 2020, suggesting minimal meltwater storage in snow/firn; c) one-day lagged air temperature displays the strongest correlation with river stage; d) river stage correlates more strongly with ablation zone albedo than with net radiation; and e) late-summer rain-on-ice events appear to trigger the region's sharpest and largest floods. The new gauging stations provide

valuable in situ hydrological observations that are freely available through the PROMICE network (https://promice.org/weather-stations/).

## Introduction


Besides solid ice discharge, climate change-induced meltwater runoff is a dominant driver of Greenland Ice Sheet (GrIS) mass loss (Mottram et al., 2019; Mouginot et al., 2019; Shepherd et al., 2020; King et al., 2020) that is projected to increase throughout the 21st century (Trusel et al., 2018; Noël et al. 2020; Hofer et al., 2020). However, current climate models typically calculate runoff as a residual term in surface mass balance budgets (van Dalum et al., 2021). As runoff represents rain and

meltwater that is not refrozen or retained in the firn, errors in the surface energy balance terms used to calculate melt/refreezing or in any of the other surface mass balance terms propagate to error in the subsequent runoff term calculation. There is therefore a growing need for accurate, in situ hydrological datasets for characterizing runoff magnitude and timing as well as evaluating ice sheet surface mass balance (SMB) models (Smith et al., 2017; Smith et al., 2021). In situ measurements of runoff are useful for direct observation of GrIS runoff contributions to sea level rise, as well as for assessing hydrological models, hydropower

potential, and freshwater resources critical for decision makers (Smith et al., 2017; Alther et al., 1981; Instanes et al., 2015).

Despite this need for in situ hydrological measurements, only a small handful of GrIS proglacial rivers have been gauged outside of SW Greenland (Ploeg et al., 2021; Mankoff et al., 2020; Mernild et al., 2008) and the majority of modelled runoff evaluation studies have been conducted in SW Greenland (Smith et al., 2017; Mernild et al., 2011; Mernild et al., 2018; Cooper

et al., 2018; Smith et al., 2015). The NW GrIS has also experienced extensive melting since 2000 and is now one of the largest contributors to GrIS mass losses (Mouginot et al., 2019). By 2100, runoff mass losses from the NW GrIS are projected to reach 201 Gt/yr, far exceeding the anticipated dynamic losses of 22 Gt/yr from the region at that time (Muntjewerf et al., 2020). However, the NW GrIS lacks in situ measurements necessary to characterize the region's hydrometeorology, quantify runoff drainage to the ocean and evaluate SMB models.


More generally, permanent river gauging installations are time- and resource-intensive to install and maintain in remote, harsh Arctic environments. Proglacial river gauging sites are typically difficult to maintain due to high sediment bedload and shifting, braided channels. In situ sensor systems generally require expert installation and repair, a difficulty compounded in remote study locations with poor accessibility. Traditional pressure-based river gauge technologies (i.e. pressure transducers (PTs))

require full immersion and are thus poorly suited for Arctic rivers where thick ice, spring floods, and rolling cobbles can cause sensor damage, movement, or loss. Recently, non-contact methods of river stage measurement including radar, laser rangefinder, and camera have received increasing interest (Bandini et al., 2022; Buyaert et al., 2020; Goldstein et al., 2023), but laser rangefinder and radar sensors are traditionally nadir-looking and thus require mounting to a bridge, precluding their

use in undeveloped Arctic locations. To our knowledge no in situ hydrological observations of this length (3+ years) are currently available for NW Greenland, an exceedingly cold and remote region.

Analysis of runoff stage data enables evaluation of diurnal, seasonal, and annual runoff patterns as well as assessment of the relationship between these patterns and meteorological drivers. However, as SMB climate models estimate runoff flux, river stage measurements alone cannot be directly compared with SMB outputs. River stage measurements must be combined with a stage-discharge curve (established with in situ discharge measurements) and careful watershed delineation to allow for comparison between in situ runoff flux and SMB climate model runoff flux. Such discharge measurements were recently collected by the author team and are currently undergoing quality control. These data will be presented with a remotely-sensed ice watershed delineation and SMB model outputs in a future publication.

This paper describes new hydrometric sensor installations, and the resulting 3-year time series (2019-2021) of river stage (water level) at three proglacial gauging sites in NW Greenland. The sensors include vented and non-vented PTs, a bubbler sensor, bank-mounted laser rangefinders, and time-lapse cameras to record river stage (water level), and an automated weather station (AWS). The bank-mounted laser rangefinders are oblique-looking, a novel approach to laser rangefinder stage measurement for remote river gauges. An initial assessment of data and instrument performance is provided for the 2019 to 2021 runoff seasons. To mitigate intermittent data gaps found in all instruments, we also create a hybrid, multi-sensor river stage product for the Minturn River, Inglefield Land. Following technical descriptions of the instruments and performance, a preliminary characterization of diurnal, seasonal, and interannual variability in NW GrIS meltwater runoff is presented. We conclude that experimental new hydrometeorological sensor installations can improve physical understanding of ice sheet runoff and SMB for a remote, little-studied, hydrologically active area of the GrIS.

## 1 Study sites

In July 2019 we established three new hydrometric sensor installations on the Minturn River (78.590801°, -68.992706°, Inglefield Land), North River (76.538980°, -68.728190° located on Thule Air Base (AB)), and Fox Canyon River (76.466460°, -68.579060°, near Thule AB) (Fig. 1). The Minturn River watershed is the largest of the three (~3,800 km$^2$, ~75% glaciated) with elevations ranging from 314 m to 1,653 m (Fig. 1, (a)). Due to differences between DEM-based watershed delineations of the Minturn, the estimate of ~3,800km$^2$ is an acceptable approximation for this paper. The smaller North River (229 km$^2$, ~56% glaciated) and Fox Canyon River (115 km$^2$, ~31% glaciated) watersheds drain through or near Thule AB (Fig. 1, (b) and (c)) with elevations ranging from 59 m to 1045 m and 199 m to 913 m, respectively. In each of the watersheds, the ice sheet is fully grounded with no known surface-to-bed connections (i.e. moulins or water-filled crevasses). Instead, ice sheet runoff is routed entirely over the ice surface through long, semi-parallel supraglacial streams directly onto bedrock-dominated proglacial zones (Fig. 2; see also Yang et al., 2019; Li et al., 2022), signifying that proglacial runoff is dominated by surface

processes with negligible influence of en- or sub-glacial hydrology. The dominance of supraglacial hydrology in NW Greenland is distinctly different from the SW GrIS, where surface runoff typically enters moulins prior to reaching the proglacial zone (Smith et al., 2015; Yang et al., 2016), sometimes even at high elevations (Gagliardini and Werder, 2018).

## 2 Instruments and data collection

### 2.1 Minturn River, Inglefield Land

The hydrometric sensor package at the Minturn River includes a compact constant flow bubbler, two custom-built, bank-mounted, oblique-looking laser rangefinder systems, an AWS, and two StarDot NetCam SC 5 MPixel cameras. The bubbler, which began measurement on July 9, 2019, is a Sutron® Constant Flow Bubbler (CF Bubbler) (Compact Constant Flow Bubbler, 2022), a self-contained, precision device to measure water level and temperature. The CF Bubbler consists of a pump, tank, manifold, control board, display/keypad within a 10.5"x 8.5"x 7.5" polycarbonate NEMA-4X enclosure. Measurements are collected every 15-minutes and transmitted via Iridium satellite modem every hour. In 2021, an additional vented Level TROLL 700H Data Logger Pressure Transducer was installed at the site. The sensor is contained in a titanium body and measures water pressure and temperature, from which water level is determined. This second vented Level TROLL PT has operated across the river channel from the CF Bubbler since June 16, 2021.

River stage measurements are complemented by two custom-built, bank-mounted, oblique-looking Laser rangefinder systems, M1 and M2 (one on each riverbank) (Fig. 3). These ruggedized systems were built at CRREL in collaboration with Crane Johnson from the National Weather Service, who previously developed their concept and programming. A Laser Technology Inc. (LTI) Trusense S200 (Buytaert and Sah, 2020), a time of flight-based long range distance finder utilizing the 905 nm wavelength, and an inclinometer are contained in an aluminum enclosure positioned with a clear line of sight to the water surface. Fifty laser returns are recorded on the hour each hour and recorded by a Campbell Scientific CR1000X data logger, along with battery voltage, panel temperature, and inclinometer angle. Hourly first, strongest, and last laser returns are transmitted with the system information and number of transmission tries via the Iridium satellite network. The system is powered by a 10-watt solar panel on a versatile polar mast mounting system, with energy stored in a deep cycle sealed lead acid battery.

Meteorological measurements are collected by an AWS which was installed at the Minturn River on July 9, 2019 (Fig. 4). Wind speed and direction are measured with a Sutron Ultrasonic Wind Sensor 5600-0215 (Wind Speed, 2022). Two Kipp and Zonen CMP3 Pyranometers (Kipp and Zonen, 2022), one facing upward and one facing downward, measure incoming and reflected solar irradiance. A Sutron Accubar Barometric Pressure Sensor 5600-0120 (Accubar, 2022) containing a highly accurate solid state pressure transducer that measures barometric pressure. Air temperature is recorded to the nearest ±0.1°C by a Sutron Air Temperature Sensor 5600-0020 (Air Temperature, 2022) and to the nearest ±0.3°C by a Sutron AT/RH High

Accuracy Sensor (AT/RH, 2022), which additionally measures relative humidity to the nearest ±1.5%. These AWS measurements are collected every 15 minutes and telemetered via Iridium satellite modem every hour.


Finally, two StarDot NetCam SC 5 MPixel cameras (Multi-Megapixel, 2019), which were installed on July 12, 2019, collect images of the Minturn River and foreground every 3 hours when there is sufficient ambient lighting (e.g. night time and winter images will be underexposed and are therefore not useful data). The timing of these image acquisitions is alternated, yielding an image every 1.5 hours. Images are transmitted to CRREL via Iridium satellite modem. Each camera view includes the

opposite bank of the river, allowing for calculation of river stage through edge detection (Goldstein et al., 2023) as well as visual evidence of river ice, rain, snow, and other environmental conditions.

During the July 2019 field campaign, Terrestrial Laser Scanner (TLS) elevation point clouds of the site were collected with a RIEGL VZ-400i 3D Laser Scanner (RIEGL, 2022), using a Trimble R10 base station and rover for GNSS positional

measurements. Scans were taken from 27 positions around the site between July 10 and July 11, 2019. Scans from each position were aggregated into a 3D digital elevation model (DEM) that includes all permanently installed instrumentation. Optical surveys were conducted from July 8 to July 11, 2019 to independently establish the elevations of instruments and river stage relative to newly established benchmarks.

Annual service trips have been completed since the instrument cluster was established in 2019. Due to international research travel restrictions to Greenland during the COVID-19 pandemic, ASIAQ Greenland Survey and Vectrus (a mission-support organization servicing Thule AB, https://www.vectrus.com/), completed the 2020 and 2021 service trips, respectively. The ASIAQ service trip on June 21, 2020 replaced the AWS batteries, corrected wiring of its solar panel, replaced its SD card, replaced a SIM card to the right bank time-lapse camera, power cycled the lasers, and repaired a broken wire on the Sutron CF

Bubbler. During a following Vectrus service trip on June 12, 2021, data was manually downloaded from all stations, the laser unit software was updated to retain the date while in sleep mode and wake up automatically on April 1, the batteries and charge controllers were all checked, the Level TROLL pressure transducer was installed, and all stations that had not yet turned on for the year were power cycled.

## 2.2 North and Fox Canyon River, Thule AB

On the right bank of the North River, a third custom-built, bank-mounted LT Trusense S200 laser rangefinder ranging system, N1, was installed during July 2019 and programmed to retrieve and telemeter river stage estimates every hour. As before, telemetry is achieved via an Iridium satellite modem. Two Solinst Levelloggers (non-vented, self-contained, self-logging PTs) were installed on the left bank of the North River in July 2019 to record stage, one inserted on a slotted steel pipe and the other inserted in a slotted PVC pipe. A single corresponding Solinst Barologger was placed in the bank-mounted laser rangefinder

box to record atmospheric pressure variations for later correction of the two PT stage records. All three pressure transducers

record data every 15 minutes, aligned with the single beam laser rangefinder. Due to the close alignment of the two non-transmitting Levelogger records, the more complete Levelogger record was identified as the preferred pressure transducer record for the North River during the time period July 17, 2019 to May 22, 2021.

A vented Level TROLL 700H Data Logger PT was installed alongside the laser rangefinder on the North River in 2021 and began transmitting records on June 4, 2021. This vented Level TROLL PT is used as the PT stage record beginning June 4, 2021. These measurements were complemented by TLS elevation point clouds of the North River which were acquired on July 5, 2019 from twelve positions with the RIEGL VZ-2000 3D Laser Scanner, Trimble R10 base station, and rover for GNSS positional measurements. Scans from three additional positions were performed on October 5 and October 9, 2019. Optical

surveys were conducted between June 30 and July 5, 2019, tying water level measurements to three newly established fixed benchmarks on boulders along the bank.

A Vectrus service trip on June 12, 2021 downloaded data files, power cycled the laser, checked the batteries and charge controllers, updated the laser software to allow for automatic spring wakeup, and installed the vented PT.


At the Fox Canyon River, a fourth custom-built, bank-mounted LT Trusense S200 laser rangefinder ranging system, F1, was installed on July 15, 2019, programmed to retrieve river stage measurements every 15 minutes and telemeter them hourly via an Iridium satellite modem. A RIEGL VZ-2000 3D Laser Scanner survey was performed during this initial installation trip to establish benchmarks. Optical surveys at the Fox Canyon River Bridge were performed between July 1 and July 4, 2019, using

the three newly established fixed benchmarks. On June 12, 2021, a Vectrus service visit downloaded data files, power cycled the laser, checked the batteries and charge controllers, and updated the laser software for spring wakeup.

### 2.3 Data telemetry, downloads, and public release

Data from all four bank-mounted laser rangefinder systems, the Minturn River CF Bubbler, the Minturn and North River Level TROLL PTs, the Minturn time-lapse cameras, and the Minturn River AWS are telemetered hourly via the Iridium SBD network

and received by the US Army Corps of Engineers. River stage and AWS measurements from the Minturn River are automatically forwarded to the Geological Survey of Denmark and Greenland (GEUS) PROMICE database (https://dataverse.geus.dk/dataverse/PROMICE) and included on their website under the station name ING_1 (PROMICE, 2023).

Non-telemetered PTs were downloaded in the field. The North River Levelogger and Barologger PTs were installed on July 17, 2019 and recorded data until May 22, 2021. Minturn River cameras began collecting data on July 12, 2019.

## 2.4 Remote sensing data

Precipitation, snow cover, and albedo were obtained using satellite sensing to characterize ice surface conditions for the GrIS ablation zone over each watershed. Due to the small size of the North River and Fox Canyon River watersheds relative to the spatial resolutions of the remote sensing datasets, these datasets are less representative of conditions within the watersheds and statistical analyses over these watersheds are omitted from this paper. Daily accumulated precipitation was derived from combined microwave IR obtained from the Integrated Multi-satellitE Retrievals for Global Precipitation Measurement (IMERG) (GPM, 2019). Snow cover data over the three watersheds was obtained from the NDSI_Snow_Cover layer and albedo was obtained from the Snow_Albedo_Daily_Tile layer from the MODIS/Terra Snow Cover Daily L3 Global 500m SIN Grid product, Version 6.1 (Data Set ID: MOD10A1) (MODIS, 2023). These remotely sensed precipitation, snow cover, and albedo data products provide insight into additional aspects of the hydrometeorology of Minturn watershed.

## 2.5 Data processing

Some minor filtering was required to remove anomalous data and outliers. Stage values from the CF Bubbler data were clipped to fall between 0.6 m, the lowest value the CF Bubbler could read based on its position in the river, and 6 m, a conservative upper estimate of river stage. At the end of the 2020 melt season (September 2 to 12, 2020) the CF Bubbler returned some anomalously high stage measurements, likely due to freezing, which were also removed. The CF Bubbler pipe was abruptly displaced downward by 0.5 m between 3:15am and 3:30am on July 29, 2021, due to turbulent flow as confirmed by the time-lapse camera imagery. Stage records following the offset were adjusted by +0.5m to maintain consistency with the rest of the record. This CF Bubbler record, along with records from all other stage instruments, was converted to water surface elevation (WSE) (WGS84/UTM Zone 19N, ellipsoid height) using optical surveys tied to fixed benchmarks.

Non-telemetered PT data at the North River were downloaded in the field using a Solinst Levelogger optical reader. The two Levelogger PT records were compensated for air pressure using the Barologger PT record and Levelogger 4.6.2 software. As the two Levelogger records were virtually identical, the one with the more complete record is used in the following analysis. Telemetered Level TROLL PT data are compensated for air pressure by CRREL. For analysis, this Level TROLL PT record was converted to WSE following the methodology described for the CF Bubbler (WGS84/UTM Zone 19N, ellipsoid height).

Telemetered returns from the bank-mounted laser rangefinder systems were received and processed into distance values by CRREL. First, last, and strongest values were transmitted to a webpage for instantaneous access. To filter outliers present in the Minturn River records, only physically realistic values were retained (between 6 m and 20 m for laser rangefinder M1 and between 5 m and 15 m for laser rangefinder M2). The median return of each record was then taken as the distance from the laser device to the water surface.

This median laser return distance was then used to calculate river stage as follows. As all rivers experienced zero flow during the cold season, the median corrected laser rangefinder return from each record was taken as the distance to the empty riverbed and/or frozen river ice surface as measured on a zero-flow day. At the Minturn River, zero-flow was visually confirmed with camera image records. Simple trigonometry was used to compute the vertical distance of the water surface below the laser box ($Z_{Lidar\ Box}$) using the measured distance to the water surface (Median Lidar Distance) and the vertical angle of the laser range finder ($\theta_{Lidar\ Box}$). This vertical distance was then used to calculate river stage (Equation 1) relative to an arbitrary datum:

$$Stage = Z_{Lidar\ Box} - Median\ Lidar\ Distance * sin(\theta_{Lidar\ Box}) \tag{1}$$

Using the TLS point cloud captured at each study location in 2019, the ellipsoid-referenced elevation of each bank-mounted laser rangefinder unit could be found in the scans, yielding the WSE of each bank-mounted laser rangefinder record (WGS84/UTM Zone 19N). Some fixed anomalous offsets, likely from solar interference, occurred occasionally in some returns and their correction is described in Appendix 1. While the snow depth instrument from the AWS failed to function, all other meteorological instruments functioned as expected and their records were transmitted to the PROMICE database without further processing.

River stage fluctuations were also extracted from time-lapse camera images by identifying the wetted shoreline position and combining this with the TLS scan of the bank topography (Goldstein, et al., 2023). Full details of this method can be found in Goldstein, et al. (2023), but, briefly, a Canny edge detector was used to automatically detect the shoreline position from each camera image, with manual delineations performed if necessary. Georectifying these shoreline positions and projecting them onto the TLS scan of the riverbank yielded estimates of river height fluctuation with an estimated error of ±0.185 m (Goldstein et al., 2023). This camera record has recognized uncertainty for the lowest ~75% of Minturn River stage values, due to mathematical extrapolation of the TLS bank scan below the waterline. For our analysis, this camera record was converted to approximate WSE by arbitrarily assigning the WSE of the CF Bubbler record to the camera record on a day of zero-flow.

Because all sensors experienced occasional data gaps, a hybrid stage product for the Minturn River was created to infill missing data (Fig. 5). Laser rangefinders M1, M2 (Minturn River), N1 (North River), and F1 (Fox Canyon River) will be referred to as Lidar M1, M2, N1, and F1, respectively, in Figures 5-7. The CF Bubbler transducer, in particular, becomes exposed to the air at water surface elevations less than ~316 m (~316.5 m following its ~0.5 m offset on July 29, 2021), with measurements from other sensors able to record lower stages. From 2019 to 2020, bank-mounted laser rangefinder M1 had the most measurements after the CF Bubbler and the two records were fit with a robust linear model (hybrid stage = 1.0647*M1 - 20.5938m supply formula and $R^2 = 0.9651$) with outliers greater than ±0.25m from the trendline removed. Beginning June 16, 2021, the additional vented Level TROLL PT provided an additional dataset with higher accuracy than the bank-mounted laser rangefinder. These hourly measurements were interpolated to 15-min measurements by fitting the Level TROLL PT record

with a linear model to fill gaps in the CF Bubbler data (hybrid stage = 0.9515*PT + 15.323m supply formula and $R^2 = 0.9981$). The bank-mounted laser rangefinder data were no longer included in the hybrid stage product after the vented Level TROLL
PT was installed, as the latter is more accurate. The final hybrid river stage product therefore uses the CF Bubbler data when available, with gaps filled with a linear regression model based on either Level TROLL PT or laser rangefinder M1 measurements as available.

## 2.6 Statistical comparisons

For quality assessment of the various experimental methods used to estimate Minturn River stage, statistical intercomparisons
were performed by calculating the $R^2$ and RMSE values of the traditional CF Bubbler vented PT with all other records (i.e. the two bank-mounted laser rangefinders, the Level TROLL PT, the time-lapse camera estimates of Goldstein et al. (2023), and the hybrid product (Table 1)). For a preliminary scientific assessment of some potential physical drivers of Minturn River stage variations, a multivariate linear regression model was developed to assess the relationship of each AWS meteorological variable to the hybrid stage product. Correlation coefficients, p-values, and scatterplots were also produced for each variable
available from the meteorological station and remote sensing data sets (Fig. A2). In these correlation analyses, AWS air temperature and remotely sensed precipitation, each lagged by 0, 1, and 2 days, were considered and net radiation was calculated as the difference between shortwave/solar downward and upward radiation. To remove high diurnal variance, all predictor variables and the hybrid river stage product were averaged to daily timesteps. Any significant, independent variables found were then used as predictor variables in a multivariate linear regression mode. These independent variables, albedo over
the ice sheet and air temperature, were used as predictors in a multivariate linear regression model predicting hybrid stage. Coefficients and p-values were computed for each predictor.

ANOVA tests were performed on the air temperature, ice sheet albedo, precipitation, net radiation, downward radiation, and hybrid stage to assess differences in means between early melt season (~day of year 190 to 205) for each year (2019, 2020,
2021). The early date selection was based on visual inspection of the runoff stage records; qualitatively, 2019 and 2020 appear to follow similar patterns of early high runoff stage and appearance of a diurnal signal, while runoff was lower and emergence of diurnal signal was delayed in 2021. To investigate these differences, we limited our early season assessment to the period between day 289 (the first day records are available in all years) and day 205 (the latest onset of a pronounced diurnal signal).

## 3 Results

### 3.1 Minturn River, Inglefield Land

Proglacial river stage measurements for the Minturn River were acquired by the CF Bubbler vented PT (15 min), two bank-mounted laser rangefinder units (hourly), and time-lapse cameras (Goldstein et al., 2023) (3 hours) in 2019, 2020, and 2021; and by the Level TROLL vented PT (hourly) in 2021 (Fig. 5). In total, data were collected between July 15, 2019 and

September 12, 2021 with prolonged winter shutdown gaps in the CF Bubbler and bank-mounted laser rangefinder units from approximately October 23, 2019 to June 6, 2020 and October 16, 2020 to June 14, 2021. After quality-assurance and filtering (see Data Processing), a total of 9,120 (15-min) CF Bubbler stage measurements, 3,093 (hourly) Level TROLL PT stage measurements, 6,461 (hourly) bank-mounted laser rangefinder M1 stage measurements, 3,608 (hourly) bank-mounted laser rangefinder M2 measurements, and 1,618 (3 hour) time-lapse camera images (Goldstein et al., 2023) were acquired for the Minturn River over the three meltwater runoff seasons.

The AWS meteorological station installed at the Minturn River collected relative humidity (28,900 measurements), relative snow depth (18,707 measurements), wind direction (27,634 measurements) and speed (27,669 measurements), upward (29,212 measurements) and downward (29,214 measurements) solar radiation, air pressure (25,223 measurements), and two measures of air temperature (29,213 measurements from each) at 15-minute increments from July 8, 2019 to September 12, 2021, with winter gaps from September 2, 2019 to June 21, 2020 and December 15, 2020 to June 12, 2021.

### 3.2 North and Fox Canyon River, Thule AB

For the North River at Thule AB, river stage measurements were recorded from 2019 to 2021 with two non-vented Levelogger PTs and one bank-mounted laser rangefinder, and with a Level TROLL vented PT beginning in 2021 (Fig. 6). The longest Levelogger PT record was retrieved in June 2022 and had collected 47,692 stage measurements from July 17, 2019 to May 22, 2021 with a cold-season gap between October 5, 2019 and April 1, 2020 and no gap in the more complete Levelogger PT record during winter 2020/2021. The PT record with no gap in the winter 2020/2021 was used for this analysis. The bank-mounted laser rangefinder N1 collected 8,696 stage measurements between July 15, 2019 and October 25, 2021 with winter gaps from October 20, 2019 to June 16, 2020 and October 1, 2020 to April 28, 2021. The transmitting vented Level TROLL PT operated from June 7, 2021 to October 25, 2021 and collected 2,919 measurements.

For the Fox Canyon River near Thule AB, 8,348 river stage estimates were acquired from bank-mounted laser rangefinder F1 between July 15, 2019 and October 31, 2021 with winter gaps from October 29, 2019 to June 29, 2020 and October 20, 2020 to June 7, 2021 (Fig. 7).

### 3.3 Sensor performances

Coefficient of determination ($R^2$) and root-mean square error (RMSE) values computed between the gold-standard CF Bubbler vented pressure transducer stage record and each of the instruments installed at the Minturn River show strong correlations (Table 1). The $R^2$ values were 0.809, 0.739, 0.996, and 0.702, respectively, for bank-mounted laser rangefinder M1, bank-mounted laser rangefinder M2, the vented PT, and the hybrid product, respectively. Corresponding RMSE values were 0.256 m, 0.251 m, 0.050 m, and 0.502 m, respectively.

Each stage measurement technology (CF Bubbler, Level TROLL PT, bank-mounted laser rangefinder, camera) had different strengths and weaknesses (Table 1). The CF Bubbler had the finest temporal resolution (15 min) and, as the technology is considered the gold standard for river stage measurement ($\pm$0.003m) (Compact Constant Flow Bubbler, 2022) its measurements were used as the study benchmark. However, the CF Bubbler installed in the Minturn River in 2019 was not submerged deeply enough to record low stages, and the instrument also missed readings at higher stages more frequently than other measurement technologies. The Level TROLL PT (RMSE = 0.050m with the CF Bubbler) took lower temporal resolution measurements (hourly) and was not established until 2021, but recorded lower stages and had fewer data gaps than the CF Bubbler (74.5% vs. 49.1% at an hourly timestep). The bank-mounted laser rangefinder units took hourly readings that aligned well with both the CF Bubbler and Level TROLL PT measurements (RMSE with the CF Bubbler: M1 = 0.256m, M2 = 0.251m). These non-contact units were safer, easier to install, and less expensive than the traditional vented PTs, but solar interference occasionally caused anomalous offsets or gaps in the records (data completeness for M1 = 77.1%, M2 = 33.9% at an hourly timestep). The time-lapse camera record matched the upward and downward movements of the CF Bubbler stage well, but had a substantially greater daily diurnal range yielding the greatest RMSE of the approaches studied (RMSE = 0.502m) and low quality at stage values in the lowest ~75% (Goldstein, et al., 2023) (Fig. 5a). The cameras also only took measurements every three hours and experienced interruptions during some adverse weather conditions (e.g. rain on the lens) of interest for stage measurement. However, the cameras were inexpensive, simple to install, experienced few data gaps (data completeness at 3-hour time step = 89.7%), and provided useful awareness of environmental conditions including river ice, snow, and rain.

As the instruments powered off during the cold season, data gaps were assessed during summer when all instruments were powered on and recording data. This period was taken to start at the first timestep in which all instruments had acquired at least one measurement, and to end on last day of stage variation (due to either ice formation or desiccation of the channel) for the winter. These dates span July 15 to September 7, 2019, June 22 to September 3, 2020, and June 15 to August 29, 2021. As laser rangefinder M2 at Minturn River did not power on until July 6, 2021 – 20 days after the all other instruments had turned on, the start of this season was taken as the first day in which all other instruments had begun recording (June 15, 2021). The Level TROLL vented PT record was only available in 2021 so its statistics were computed from 2021 data only. Data completeness was calculated as the ratio of number of measurements to the number of 15 min (CF Bubbler), hourly (bank-mounted laser rangefinder, vented PT), or 3-hourly (camera) time periods between the start and end times each year. From these calculations, the time-lapse camera yielded the most complete data record (89.7%) whereas laser rangefinder M2 and the CF Bubbler yielded the least complete records (33.9% and 44.4%, respectively) (Table 1).

The described differences in sensor data record completeness and accuracy encouraged the development of a hybrid product offering a more complete time-series than any individual record (Fig. 5). This hybrid product, which merges stage estimates from the CF Bubbler, bank-mounted laser rangefinder (2019 and 2020), and vented Level TROLL PT (2021), yields the most

complete and most accurate product, compared to the CF Bubbler standard (Table 1). As the CF Bubbler measurements were used whenever they were available, the hybrid and bubbler record matched exactly at all timesteps with bubbler records available. Overall, the hybrid product yielded the record with the fewest gaps at the 15 min (57.1%), hourly (89.1%), or daily (99.0%) timescales. River stage from all instruments are presented for the Minturn River, North River, and Fox Canyon River in Figs. 5, 6, and 7, respectively.

### 3.4 Watershed characterization

The acquired stage records from the Minturn River, North River, and Fox Canyon River allow a preliminary assessment of runoff characteristics in the Inglefield Land and Thule regions. Weather conditions at the Minturn River AWS were similar between years, with air temperatures reaching highs of ~15 ºC in July and declining thereafter, aligning with the July peak and

subsequent decline in net solar radiation (Fig. 8). At all sites, river levels were high in early summer 2019 and 2020, but decreased throughout the summer and ceased flowing by September. In 2019 and 2020 the diurnal cycle of river stage was well developed by July, suggesting runoff from bare ice. This was not the case in 2021, when July diurnal cycles were suppressed at all three sites (Fig. 5). In all years, late-season rain events, as verified with camera images and decreased downward solar irradiance, triggered some sharp, late-season floods in the Minturn River (Fig. 5, 2019 and 2020), North River

(Fig. 6, 2019 and 2020), and Fox Canyon River (Fig. 7, 2019).

After applying a conservative Bonferroni correction, air temperature (at all lag steps), upward/downward/net solar radiation, ice sheet/proglacial/full watershed albedo, and ice sheet/full watershed snow cover were significantly correlated with hybrid stage (Fig. A2). As air temperature at all lag steps (0-2 day) showed similar, strong correlation and significance values, 2-day

average (lag 0-1 day) and 3-day average (lag 0-2 day) air temperature variables were introduced. The strongest predictor from each group of correlated variables was selected. This yielded 2-day average air temperature (p-value=$1.17 \times 10^{-62}$, $R^2$=0.650) and ablation zone albedo (p-value=$4.10 \times 10^{-71}$, $R^2$=0.620) as the independent predictors included in the multivariate linear regression model for hybrid stage prediction (Fig. A2). P-values for correlation with stage in the multivariate linear regression model were $1.38 \times 10^{-13}$ and $5.73 \times 10^{-7}$ for temperature and albedo, respectively. The linear

model coefficients were 0.077 for 2-day average air temperature and -0.030 for ice sheet albedo. Regressions of hybrid stage with remotely sensed precipitation at 0- to 2-day lags, air pressure, and proglacial snow cover showed no independent correlation (p-values $8.97 \times 10^{-3}$ to 0.931, $\alpha$=$2.78 \times 10^{-3}$). We conclude that air temperature and ablation zone albedo are the primary drivers of runoff stage for the Minturn watershed across seasons and years.

An early melt season analysis of variance test on the yearly temperature, albedo, remotely sensed precipitation, net radiation,

downward radiation, and stage data from day 189 (July 7/8, the first day a stage record is available in all years) to day 205 (July 23/24, the latest onset of a pronounced diurnal signal) indicates significant differences were present between 2021 and

2019/2020 for temperature, albedo, and stage. No significant differences existed between the means of remotely sensed precipitation, net radiation, or downward radiation in any years. Late June temperature and albedo were significantly cooler (11.2 and 10.1°C in 2019 and 2020, 6.3°C in 2021), had a more reflective ablation zone surface (0.64 and 0.65 in 2019 and 2020, 0.75 in 2021), and yielded lower river stages (317.2m and 316.9m in 2019 and 2020, 315.7m in 2021) in 2021 than either previous year. The p-value of difference in means is $1.17 \times 10^{-8}$ for temperature, $4.70 \times 10^{-4}$ for albedo, and $2.48 \times 10^{-15}$ for stage. This suggests that differences in air temperature and ablation zone albedo may have driven the differences in early season stage between 2019/2020 and 2021.

The 2-day ($R^2=0.650$, p-value=$1.17 \times 10^{-62}$) and 3-day ($R^2=0.649$, p-value=$7.15 \times 10^{-62}$) average temperatures were more highly correlated with stage than any of the single day measurements of air temperature (0-day lag $R^2=0.620$, p-value=$1.33 \times 10^{-58}$; 1-day lag $R^2=0.622$, p-value=$1.47 \times 10^{-58}$; 2-day lag $R^2=0.599$, p-value=$1.11 \times 10^{-54}$). This indicates that periods of prolonged high temperature have a greater influence on stage than single day temperatures. The similarity in the significance and correlation between 0-day and 1-day lag air temperature, with decreased significance and correlation for 2-day lag air temperature, may reflect that it takes less than two days for high temperatures to produce melt and for excess meltwater runoff to be routed from across the watershed to the gauge. This conclusion is physically realistic given the size (~3,800 km$^2$, 75% glaciated) and roughly square geometry of the Minturn watershed.

The selection of multivariate regression predictors and the ANOVA analysis of the early season meteorological differences both affirm that air temperature and ice sheet albedo are primary drivers of runoff production over the Minturn watershed. Downward radiation ($R^2=0.311$ p-value=$1.41 \times 10^{-23}$) was a more significant driver than upward solar radiation ($R^2=0.161$, p-value=$6.32 \times 10^{-12}$), which could potentially reflect that decreased cloudiness leads to more meltwater runoff generation. However, albedo over all regions (ice sheet ablation zone, proglacial zone, and full watershed) showed more significant relationships with stage, particularly ice sheet ablation zone albedo ($R^2=0.620$, p-value=$4.10 \times 10^{-71}$). Given the correlation between radiation variables and albedo variables, we conclude that air temperature and ice sheet ablation zone albedo are the preeminent drivers of stage. This conclusion is supported by the early season ANOVA analysis, which found significant differences in ice sheet albedo and stage, but not net radiation or downward radiation, between 2019/2020 and 2021.

Precipitation was not a significant predictor of stage in the linear regression model, utilizing the full three years of data, nor in the early season ANOVA analysis. This may be due to the coarse resolution (0.1° x 0.1°) of IMERG daily accumulated precipitation dataset. The limited correlation with precipitation may also be due to the infrequent and relatively small precipitation events in this high Arctic environment. Precipitation events seen in the camera imagery preceded the major floods seen in each year, suggesting that rain-on-snow events may influence the magnitude of runoff events but not be driving factors under typical (non-precipitation) conditions during the majority of the melt season.

Overall, we attribute lower proglacial river stages observed in 2021 to cooler air temperature and higher ice sheet albedo in the early part of the melt season. In the late runoff season, rainfall-induced floods occurred at all sites, confirmed by the presence of rain in the time lapse camera images and increased stage in the Minturn River. These findings affirm the influence of rainfall/melt events in NW Greenland, as previously demonstrated for SW Greenland by Doyle et al., 2015 (Doyle et al., 2015).

### 3.5 Limitations

While our installation of three proglacial river gauging stations significantly advances in situ hydrological monitoring of NW Greenland, we experienced numerous technical and logistical challenges. Submerged sensors, in particular, remain vulnerable to damage. The AWS was battery-power limited while the bank-mounted laser rangefinder devices were limited by datalogger date retention. The bank-mounted laser rangefinders were programmed to enter sleep mode from October 31 to April 1, but an error in the datalogger program resulted in a failure to properly retain the current date in the memory which prevented the datalogger from waking up in April 2020 and 2021. Due to COVID-19 pandemic travel bans, these issues require virtual training and emergency and service visits from Greenland-based collaborators. Installing multiple sensor technologies for river stage measurements provided increased resilience to these challenges, and the opportunity to create a hybrid stage product as demonstrated here. All repairs have been completed and all sensors are currently functional and transmitting data to the PROMICE website at this time.

### Conclusion

We have established three new proglacial river gauging stations and one automated weather station in the NW Greenland region. The new stations provide valuable in situ observations for characterizing the timing and drivers of GrIS runoff. Following release of the stage-discharge curves established for the Minturn and North Rivers, these data can be used to test SMB models used to predict surface mass loss from runoff in an understudied, rapidly changing area. Instruments installed at the Minturn River, Inglefield Land consist of a conventional vented pressure transducer, a bubbler sensor, and two experimental bank-mounted laser rangefinders to estimate stage, two time-lapse cameras to image environmental conditions and estimate stage, and an automated weather station to record meteorological variables (upward/downward shortwave/solar radiation, air pressure, air temperature, relative humidity, wind speed, wind direction). At the North River, Thule AB, instruments consist of one conventional vented pressure transducer, one experimental bank-mounted laser rangefinder, and two non-vented pressure transducers to measure stage. At Fox Canyon River, near Thule AB, there is a bank-mounted laser rangefinder to estimate stage. Water surface elevations at all sites are referenced to benchmarks established by this project in 2019. All instruments provided reasonably accurate, useful observations of river stage. Telemetry of these data was successful

from all sites despite harsh environmental conditions. Redundant measurements from a suite of instruments with differing

strengths and weaknesses enabled creation of a hybrid river stage product for the Minturn River with minimal data gaps. Air temperature and ice ablation zone albedo are strongly correlated with this product, providing insight into leading drivers of proglacial runoff in NW Greenland. All datasets are maintained and freely available through the PROMICE observational network (PROMICE, 2022).

**Code and data availability**

Inglefield data are available through the PROMICE network (station: ING_1). All data from all stations and corresponding code will be publicly released through Zenodo following the acceptance of this manuscript.


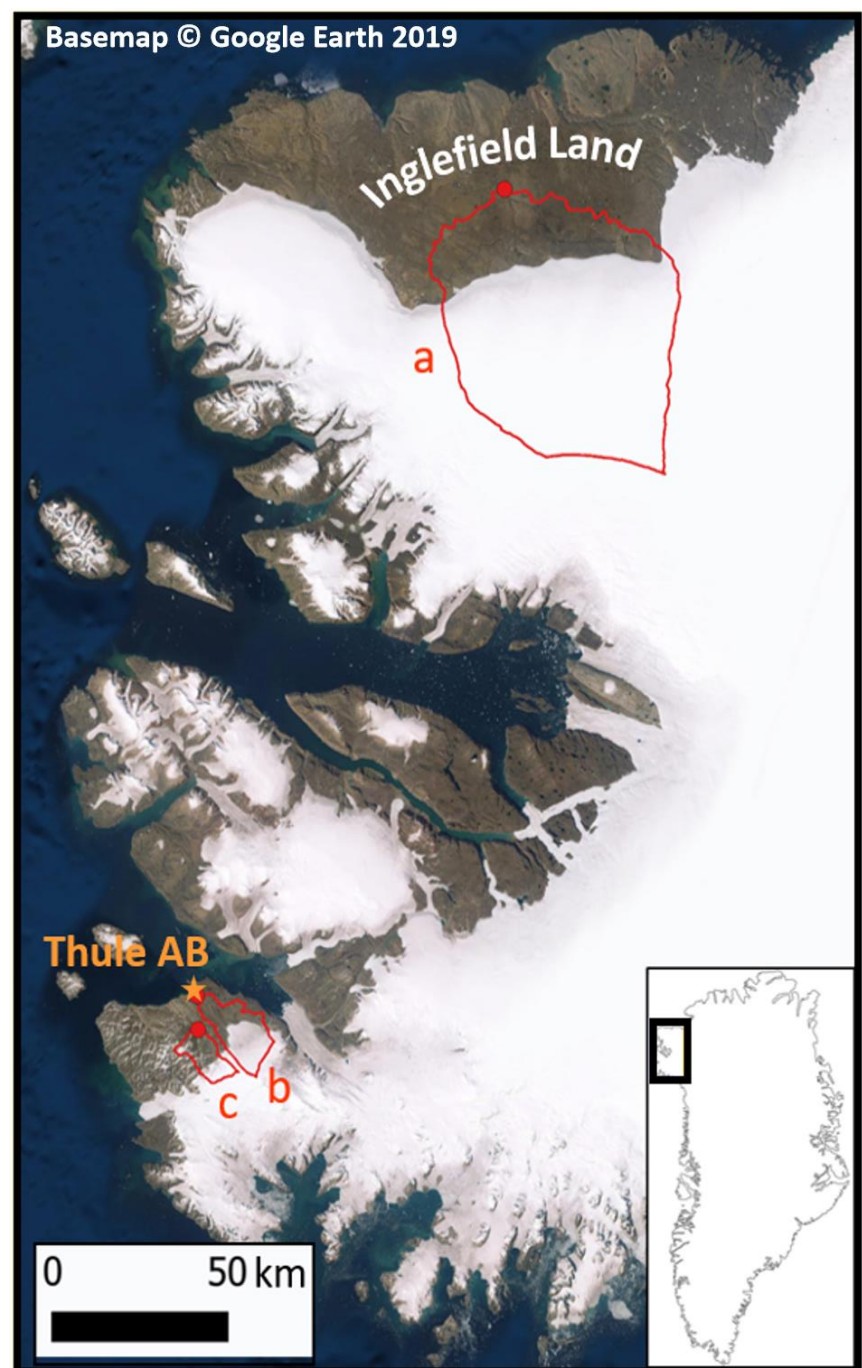

**Figure 1: New river stage measurements are available for three glacial/proglacial watersheds in NW Greenland: (a) Minturn River; (b) North River; and (c) Fox Canyon River. Instrument installations were established at (78.590801°, -68.992706°), (76.538980°, -68.728190°), and (76.466460°, -68.579060°), respectively, in July 2019 (red dots). The most comprehensive suite of sensors, including**

an automated weather station (AWS), is installed at the Minturn River, Inglefield Land (a). Thule Air Force Base, which collects airport meteorological records, is indicated by the star. Basemap © Google Earth 2019.

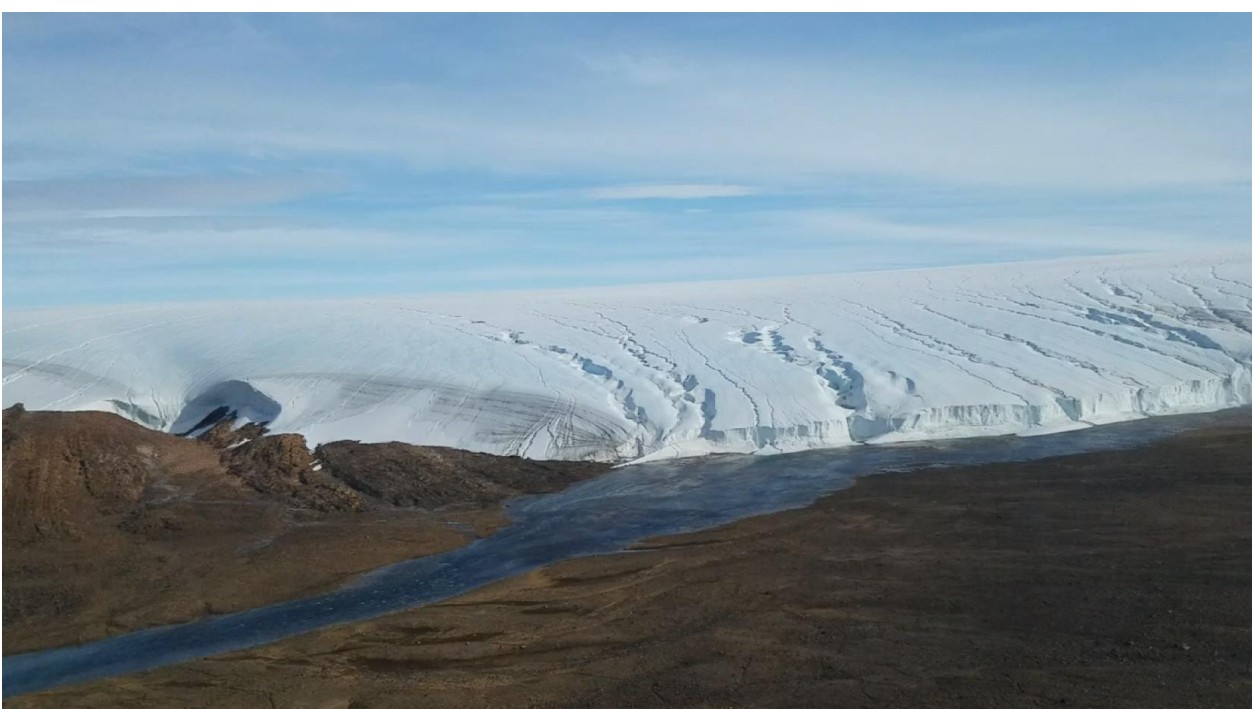

**Figure 2. GrIS runoff in Inglefield Land is routed entirely over the ice sheet surface through long, semi-parallel supraglacial streams**
**directly onto the bedrock-dominated proglacial zone, making it an ideal location to study surface runoff without interference from en- or sub-glacial processes. The absence of bedload and suspended sediment in the Minturn River at the ice margin further affirms the primacy of supraglacial runoff at this site (photo by Laurence C. Smith).**

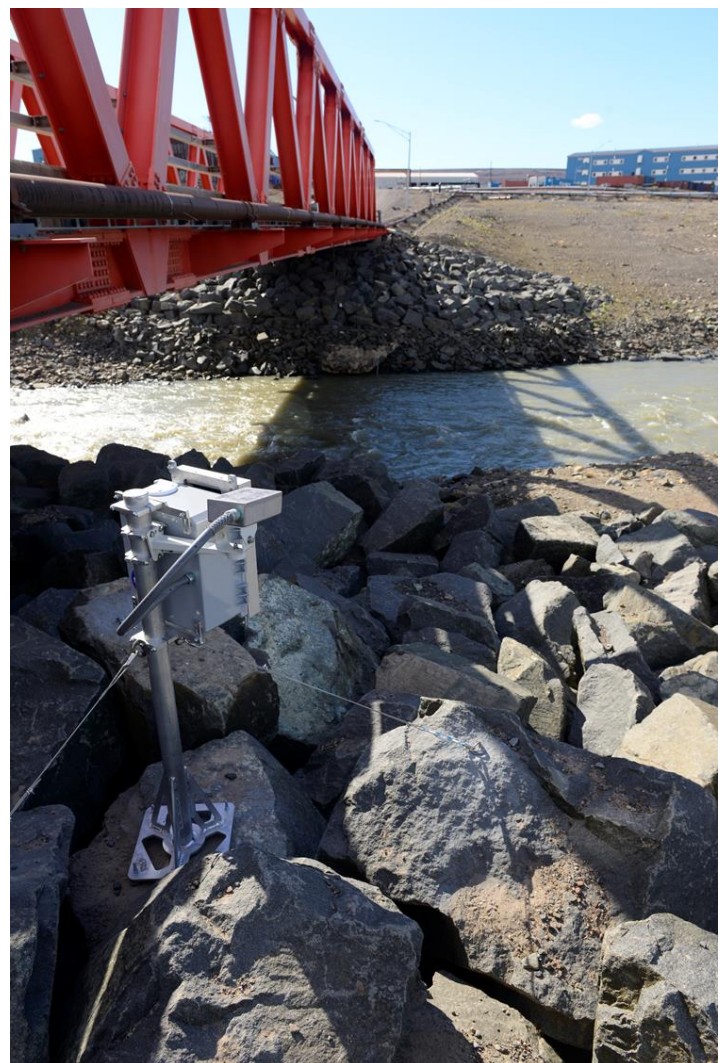

**Figure 3: Bank-mounted, oblique-looking Laser rangefinder recording river stage on the right bank of the North River, Thule AB. In total, four identical Laser rangefinder units were built and installed in NW Greenland, with two at the Minturn River (Inglefield Land), one at the North River (Thule AB), and one at Fox Canyon River (Thule AB). The single-shot Laser rangefinder measures distance between the sensor and water surface using time-of-flight of the emitted and reflected beam. This novel approach to river stage measurement avoids contact with water, reducing risk of sensor damage or loss from ice, rolling cobbles, and spring breakup floods (photo by Lincoln Pitcher).**

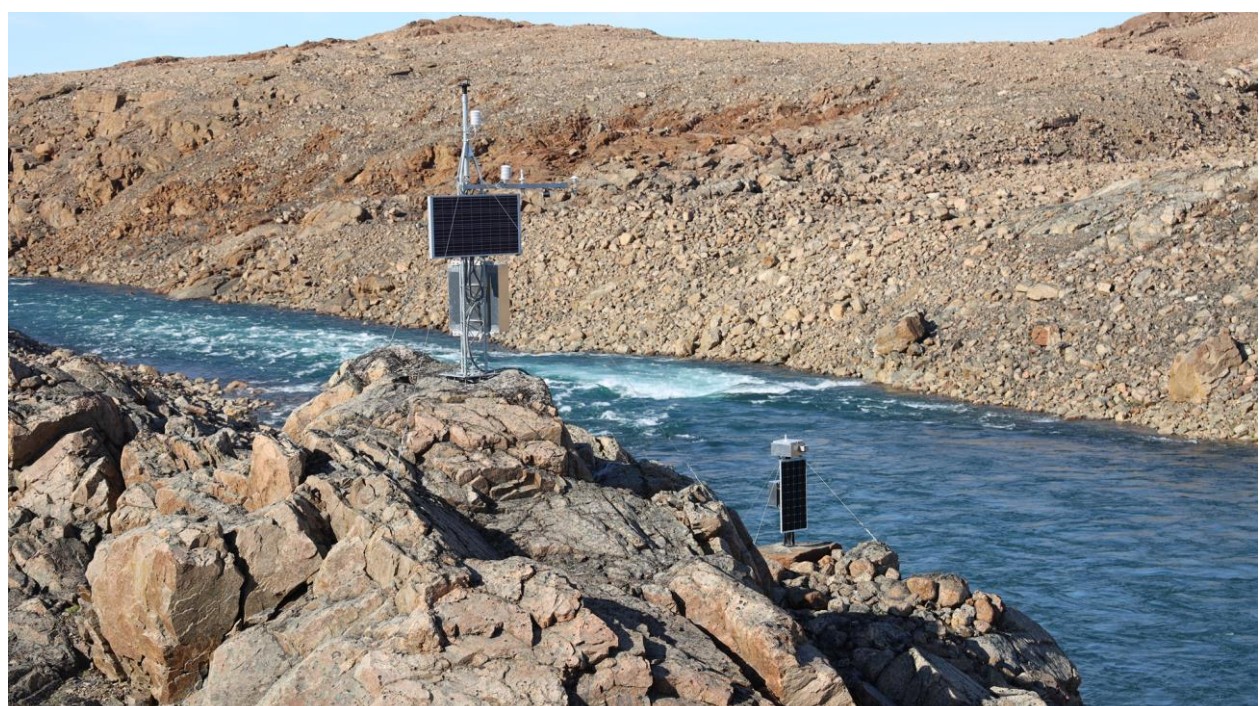

**Figure 4: Automated weather station and CF Bubbler vented pressure transducer (at left) and a time-lapse camera (at right) established at the Minturn River. Other instruments at this site include two bank-mounted laser rangefinders, a second vented pressure transducer, and a second time-lapse camera on the opposite bank (photo by Lincoln Pitcher).**


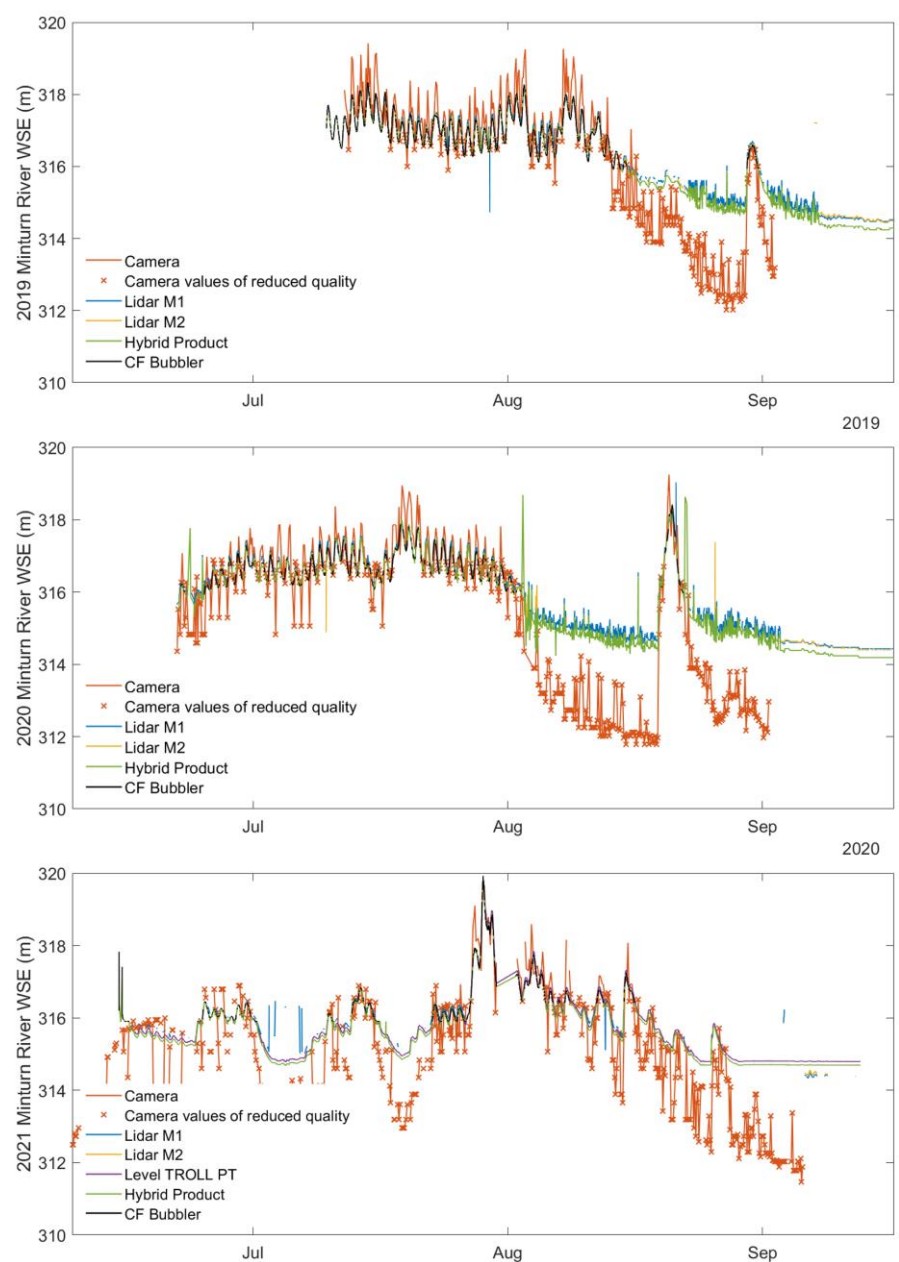

**Figure 5: Hydrographs of stage at the Minturn River (a) were produced by two bank-mounted laser rangefinder devices (Lidar M1 and Lidar M2), the CF Bubbler, and a hybrid product combination of the two from 2019 through 2021. In 2021, an additional vented Level TROLL PT was added at the Minturn River providing another record of stage. The CF Bubbler record is the gold standard for accuracy but does not capture low stages, while the lower temporal resolution, more novel laser rangefinder record may have lower accuracy. A hybrid product fit a linear model to the CF Bubbler and the bank-mounted laser rangefinder with fewer data gaps to produce the most complete record. Camera stage values from the lowest ~75% of the riverbank, denoted with X markers, contain uncertainty due to mathematical extrapolation of the TLS bank scan below the waterline.**

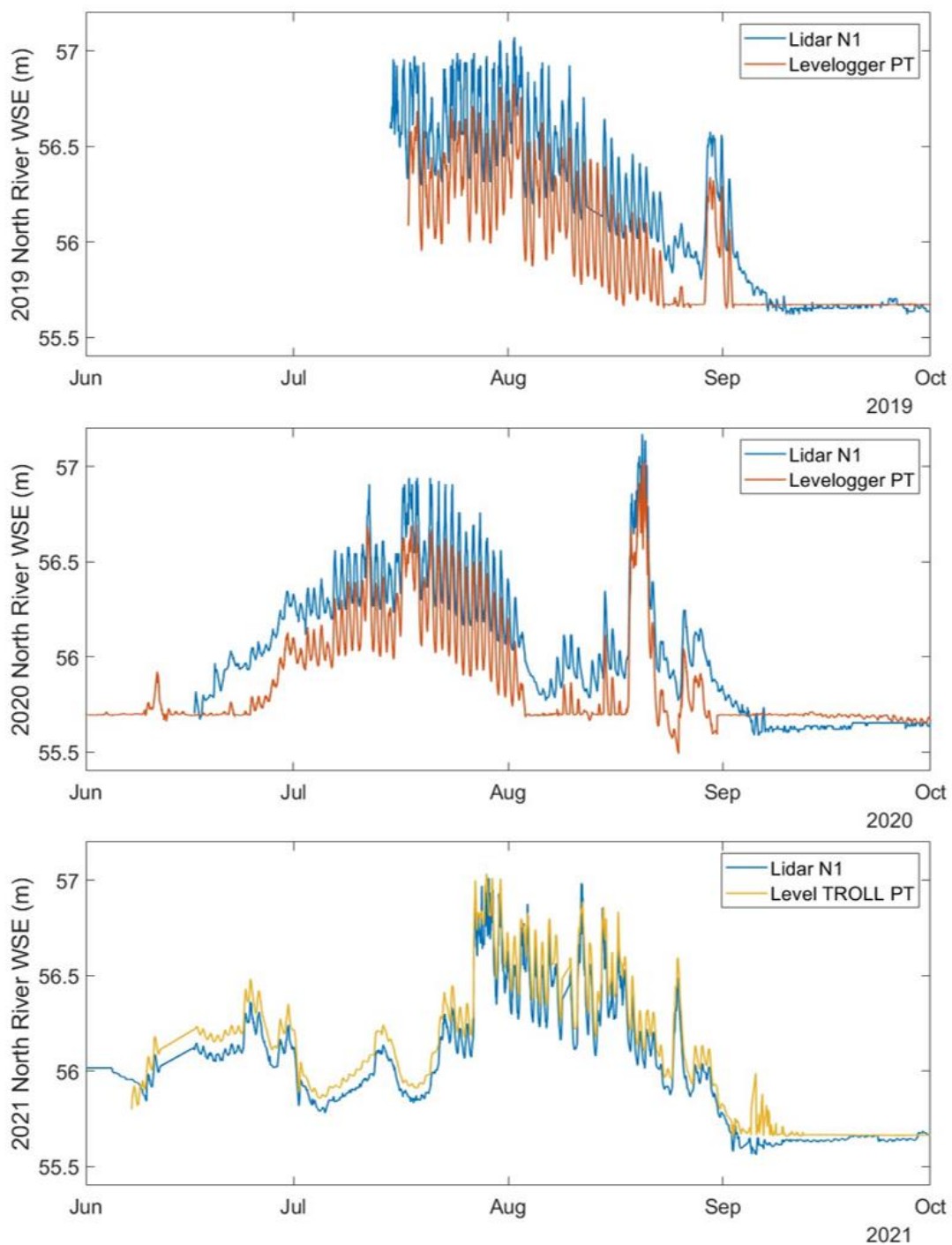

**Figure 6. Records of stage at the North River are available from a bank-mounted laser rangefinder unit (Lidar N1), with an additional record of stage available from the non-vented Levelogger (2019 to 2020) and vented Level TROLL (2021) PTs.**


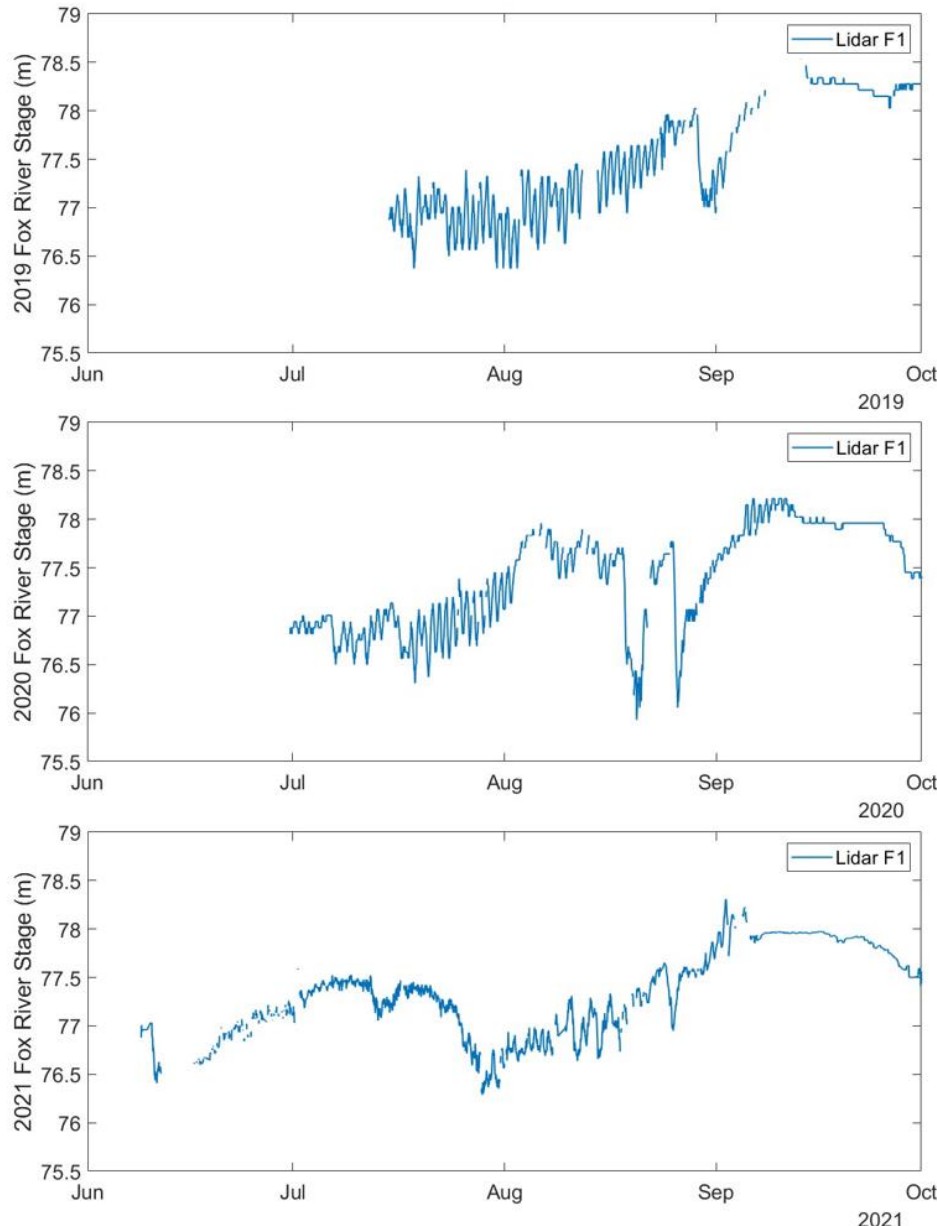

**Figure 7. Stage records from 2019 to 2021 at the Fox Canyon River are available from a bank mounted laser rangefinder unit (Lidar F1).**

**Table 1. Coefficients of determination ($R^2$) and root-mean-square-errors (RMSE) of different river stage sensors relative to the CF Bubbler vented pressure transducer record (considered the gold standard for stage accuracy, $\pm$0.003m).[20] Sensor pros/cons, including data completeness at 15-min, hourly, and daily timesteps, are summarized.**

| Dataset (measurement timestep) | $R^2$ with CF Bubbler | RMSE with CF Bubbler (m) | Data completeness (instrument timestep) | Data completeness | | | Pros | Cons |
|---|---|---|---|---|---|---|---|---|
| | | | | 15-min | Hour | Day | | |
| CF Bubbler (15 min) | N/A | N/A | 44.4% | 44.4% | 49.1% | 56.6% | Highly accurate, well established technology | Some data intermittency, requires expert installation, contact method susceptible to river ice damage |
| Bank-mounted laser rangefinder (1 hr) | M1: 0.809 M2: 0.739 | M1: 0.256 M2: 0.251 | M1: 77.1% M2: 33.9% | M1: 19.3% M2: 8.5% | M1: 77.1% M2: 33.9% | M1: 98.5% M2: 62.9% | Moderately accurate, non-contact method, simpler installation | Anomalies when sun is low, some data intermittency |
| Vented Level TROLL Pressure Transducer (1 hr) | 0.996 | 0.050 | 74.5% | 18.6% | 74.5% | 96.1% | Highly accurate, well established technology | Requires expert installation, contact method susceptible to river ice damage |
| Time-lapse camera (3 hr) | 0.702 | 0.502 | 89.7% | 7.5% | 27.2% | 95.6% | Non-contact method, few data gaps, simplest installation | Lower accuracy (especially in diurnal range and below TLS waterline), lower temporal resolution (3 hr), gaps during some inclement weather periods |
| Hybrid Product (15 min) | 1.000* | 0.000* | 57.1% | 57.1% | 89.1% | 99.0% | Most complete stage record | Derivative, multi-sensor product with variable accuracy depending on sensor |

*Note: Hybrid stage product uses CF Bubbler data when it is available, so these records match perfectly and are assumed to yield no error beyond the nominal sensor accuracy ($\pm$0.003m).[20]

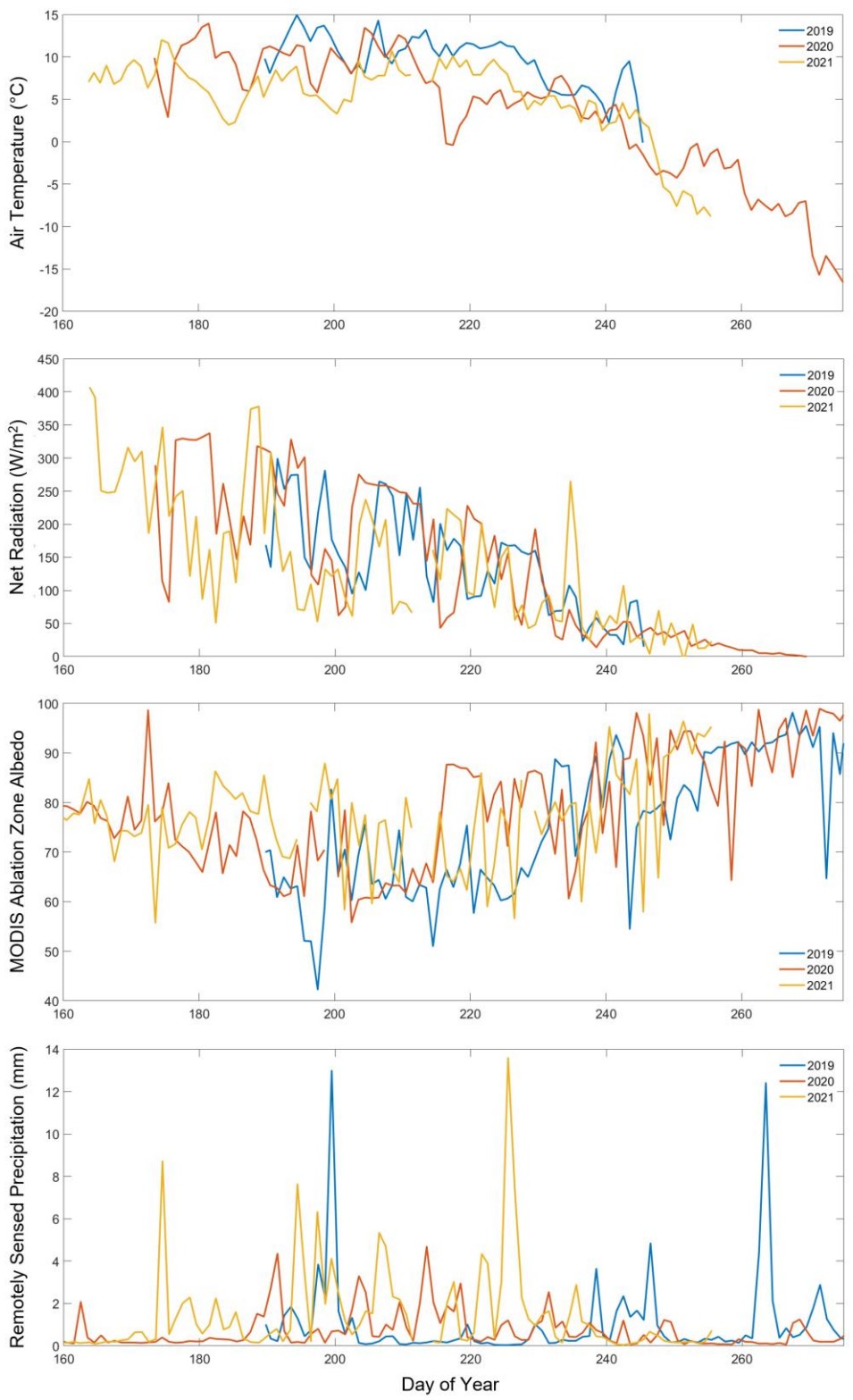

**Figure 8. Daily-averaged a) air temperature and b) net shortwave/solar radiation from the Minturn River automated weather station (AWS); and remotely sensed c) ice ablation zone albedo and d) precipitation from MODIS and IMERG, respectively for d.o.y. 160 - 275.**

**Appendix A**

Anomalous, internally consistent offsets occurred in some laser rangefinder returns between August and September of all years. In the late summer, data values were offset below the main trendline, especially between approximately 12:00pm and 12:00am, by one of two values (Fig. A1). These two offset values (-6.00m and -9.00m) were approximately constant across lasers at all locations, and the consistency of the offset timing suggests that the anomaly is a product of solar interference with the laser range finder. This explanation is corroborated by camera images of the Minturn River taken every three hours which

show strong glint on the water during the pm hours in the late summer. As some values recorded between 12:00am and 12:00pm aligned with the trendline before and after the period when the anomalies were present, this subset of the data series was taken as true values. This data series was corrected by adding one of these two constant offset values to the falsely low values (Fig. A1).

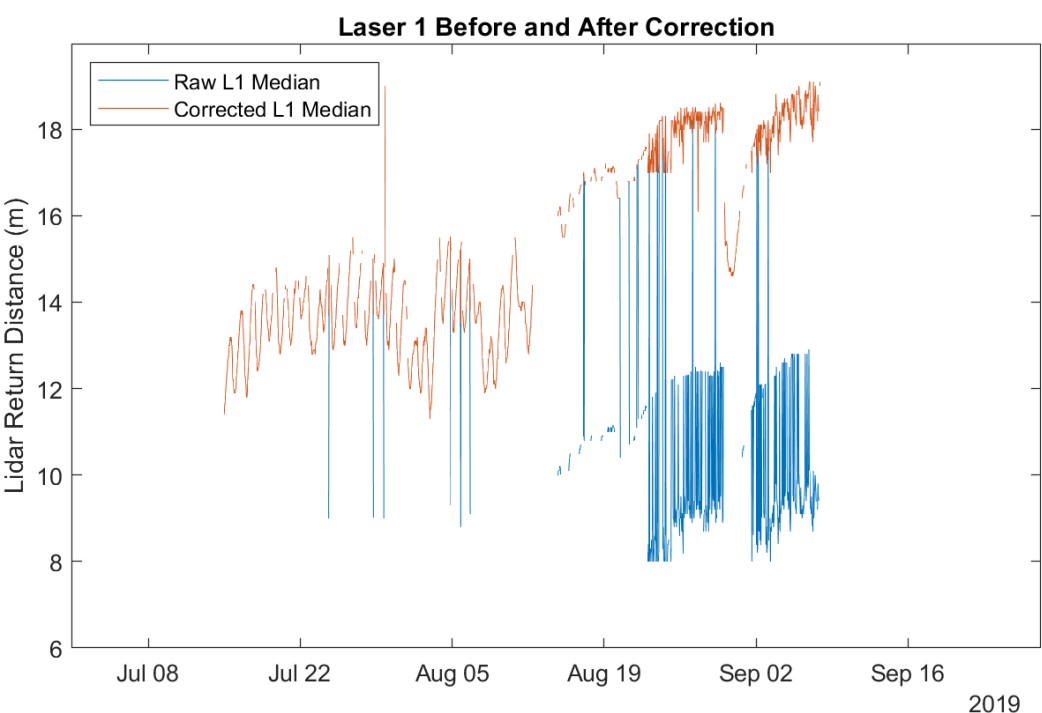


**Figure A1. Two anomalous, consistent offsets were observed in the laser rangefinder returns between August and September of both 2019 and 2020, creating an upper (primary), middle, and lower trendline in the raw (blue) data. The consistency of these offsets across lasers at different locations, the presence of the offsets in only the pm hours of the late summer, and camera images indicating strong glint off the Minturn River during these readings suggest that solar**

**interference with the laser range finder was the source of the anomalies. As both offsets adjusted the data by a constant**

**value below the primary trendline, the data were corrected by -6m or -9m produce the corrected (red) record. Similar corrections were performed for all bank-mounted laser rangefinder records.**

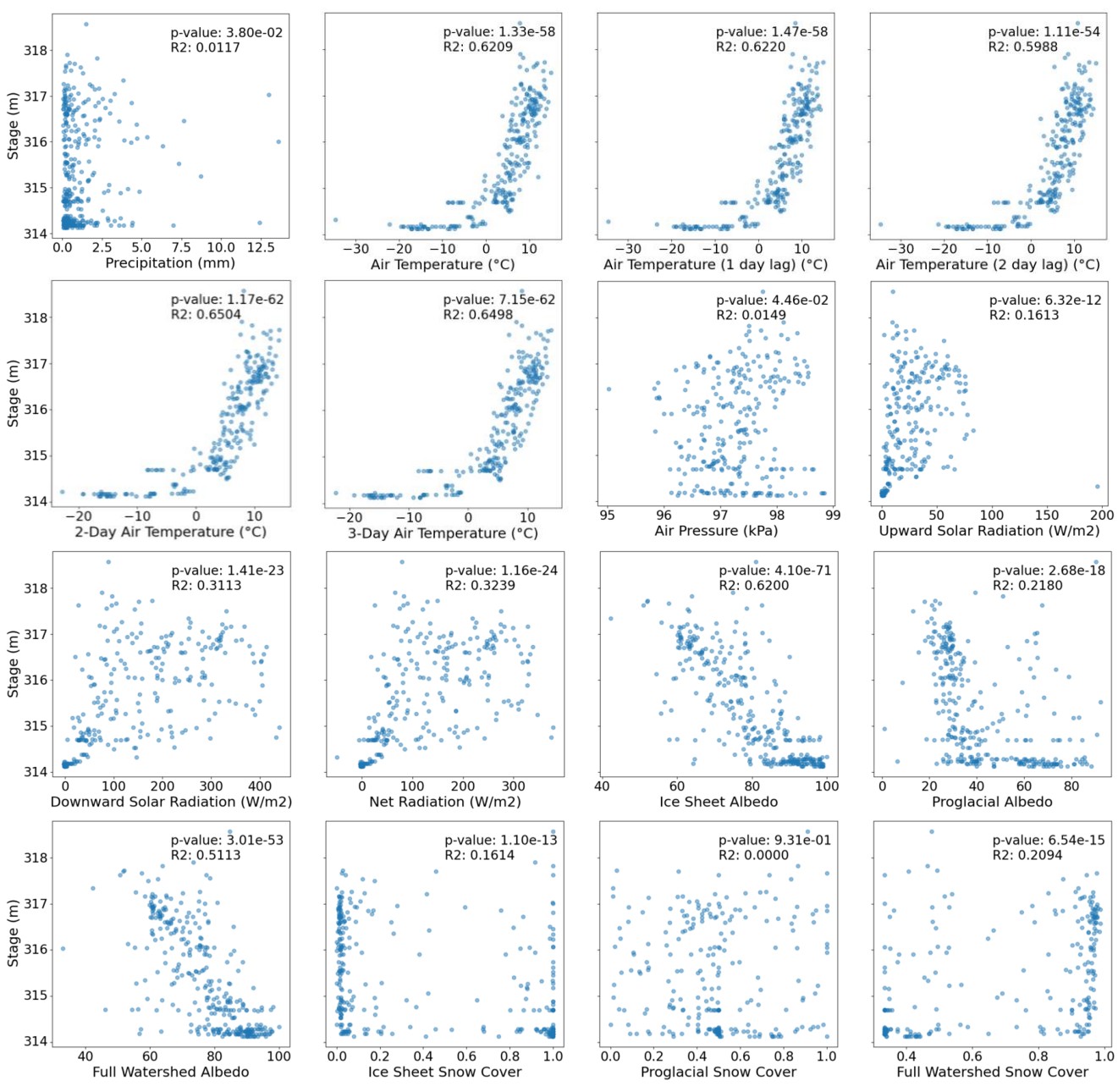


**Figure A2. Scatterplots of meteorological variables with hybrid stage. The similarity of the significance of 0- and 1-day lagged air temperature led to the inclusion of 2-day average (0-day lag and 1-day lag) and 3-day average (0-, 1-, and 2-**

**day lag) air temperature. After applying a conservative Bonferroni correction to the significance threshold and selecting the variable with the strongest significance from clusters of variables, 2-day average air temperature and ice sheet (ablation zone) albedo were the significant independent predictors selected for the multivariate regression model.**

**Author contribution**

S.E. Esenther cleaned the data, designed and carried out the analyses. L.C. Smith obtained funding and conceived and designed the study with major contributions from A. LeWinter and L.H. Pitcher. A. LeWinter, L.H. Pitcher, B.T. Overstreet, and C. Onclin carried out the fieldwork. A. LeWinter and A. Kehl designed the instruments and oversaw data telemetry and release through PROMICE. S. Goldstein and J.C. Ryan extracted stage records from the time lapse camera record. S.E. Esenther
prepared the manuscript with contributions from all co-authors.

**Competing interests**

The authors declare that they have no conflict of interest

**Acknowledgements**

This research was funded by the NASA Cryospheric Sciences program (grant #80NSSC19K0942), managed by Dr. Thorsten
Markus. We gratefully acknowledge Crane Johnson at NOAA for his single beam rangefinder designs, code, and guidance, and ASIAQ – Greenland Survey and Vectrus for facilitating service visits during pandemic travel restrictions.

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
