# Peer review of "New Proglacial Meteorology and River Stage Observations from Inglefield Land and Thule, NW Greenland"

_Geoscientific Instrumentation, Methods and Data Systems, 2023_

## Referee Comment (RC1)

**Review** *"New Hydrometeorological Observations from Inglefield Land and Thule, NW Greenland"* by Esenther et al.

The authors present new field instruments to measure river stage (water level) at three locations in northwest Greenland over 2019-2021. Instruments include a CF Bubbler, laser rangefinder units, pressure transducers and time-lapse cameras, recording stage values at different temporal resolution (15-min, hourly, 3-hourly) with various uncertainties and data gaps. To address this, the authors introduce a hybrid product that merges the most accurate sensor estimates from the extensively monitored Minturn River location. The authors highlight the added value of the hybrid product which yields the highest accuracy, while having the shortest data gaps among all other products. Using remote sensing (precipitation and albedo) and meteorological data (shortwave radiations, air temperature) from an automatic weather station installed at Minturn River, the authors identify the drivers of river stage changes.

I enjoyed reading the paper, which is very well written, and thoroughly describes the instruments used to estimate river stage, including uncertainties, pros and cons of each technique. The presented data set will undoubtedly be of high interest to the community. However, I missed a more elaborate analysis on the potential drivers of river stage variability in section 3.4. I also have some concerns about the "suggested purpose" of this new data set, i.e., whether it can be used for climate model evaluation. Below, the authors will find my General, Point and Stylistic comments. Based on these, I recommend **major revisions** before acceptance.

**General comments**

1. The abstract, introduction and conclusions highlight the need for new surface runoff estimates to evaluate (SMB) climate models. However, the actual river stage as measured in this study is not simulated by climate models, which estimate runoff fluxes, preventing a direct comparison. If possible, the authors could attempt to estimate runoff fluxes combining the hybrid stage product with the riverbed level and section. The resulting runoff flux data set could then be directly used for climate model evaluation. If this cannot be achieved with available measurements, I recommend reformulating the statements about model evaluation (e.g., L17-19, L39-41, L45-46, L50-51, L71-73, L408-409), and explicitly state that river stage is measured, but not the runoff flux, to avoid confusions. The manuscript title could be reformulated as: "River stage observations from …"

2. The main results section 3.4 is relatively short. The correlation analysis between the hybrid product and meteorological variables to identify drivers of river stage variability is interesting. However, no figures support the analysis, making it hard for the reader to interpret the results. The authors could consider adding scatterplots between each tested meteorological variables (x-axis) and the hybrid stage product (y-axis) and provide corresponding $R^2$ and p-values within each graph. The authors could also elaborate on the impact of e.g., precipitation (rain), air temperature, cloudiness and shortwave radiation, ice albedo on the recorded stage variability at Minturn River. This is initiated in L345-353 and L363-366, but a broader analysis would be beneficial.

3. The discussion section currently repeats some information presented in the results section (e.g., L368-386), suggesting that these sections could be merged.

**Point comments**

**L36:** Runoff and solid ice discharge are the two major contributors to GrIS mass loss, I suggest reformulating as: "Besides solid ice discharge, climate change-induced meltwater runoff is a dominant driver of Greenland ice sheet (GrIS) mass loss that is projected to increase …" or something equivalent. The authors could consider providing more recent references including e.g., Mouginot et al. (2019), King et al. (2020) on the contributors to GrIS mass loss, and Trusel et al. (2018), Noël et al. (2020) or Hofer et al. (2020) on the meltwater runoff increase in the 21st century.

**L38-39:** What do you mean by mass budget residual? In climate models, runoff is estimated using tipping bucket snow models that represent surface melt (solving the surface energy budget (SEB)), meltwater retention and refreezing in firn layers, and subsequent runoff. Could you clarify the statement?

**L39-41:** I am confused here, as the river stage measurements cannot be directly compared to modeled runoff fluxes from climate models. See also General comment #1.

**L45-47:** Mankoff et al. (2020) compared modeled and measured runoff from different locations, including Qaanaaq in NW Greenland and Zackenberg in NE Greenland. Please, clarify.

**L61-62:** Runoff discharge measurements are available from Qaanaaq in NW Greenland in Mankoff et al. (2020). Please, clarify.

**L123:** I am not sure about the journal policy on unpublished references. Either remove or use "personal communication". The same holds across the manuscript.

**L182-183:** Do you mean that the two other watersheds are too small to derive significant statistics? Please, clarify.

**L209-210:** For clarity, the authors could state that laser rangefinder M1, M2, N1 and F1 located in the three regions will be referred to as Lidar M1, M2, N1 and F1 in Figures 5-7. This should be done after introducing each of these sensors for the first time.

**L216-219:** Could you refer to the terms of equation 1 in the main text: e.g. "surface below the laser box ($Z_{Lidar\ Box}$) ... distance to water surface (Median Lidar Distance) ... vertical angle of the laser range finder ($\theta_{Lidar\ Box}$)"

**L235:** Could you highlight these lowest 75% stage values in Fig. 5 e.g., using colored shades? The reader can thus directly notice when the camera data set has the largest uncertainty.

**L246:** What do you mean by "predict CF Bubbler"? Fill the gap in that data set?

**L255-265:** The assessment described here is not supported by any figure/table and is only briefly discussed in section 3.4. See also General comment #2.

**L289:** Do you mean "July 15" instead of "June 15"?

**L343:** In section 3.4, could you refer to Figures 5-7 where appropriate, and explicitly state the range of discussed "days of the year".

**L345:** The authors should state that the AWS data are only available at Minturn River.

**L348:** "diurnal cycle of river stage"?

**Style**
**L20:** "SMB model evaluation" instead of "validation". L40: "evaluating", L46: "evaluation" and L51: "evaluate".
**L32-33:** For conciseness: "hydrological observations that are freely available ..."
**L45:** What do you mean by "SMB runoff", "modeled runoff"? Please, clarify.
**L50:** "quantify runoff drainage to the ocean and evaluate SMB models."
**L60:** "thus require"
**L64:** Maybe: "This paper describes new hydrometric sensor installations, and the resulting 3-year time series (2019-2021) of river stage (water level) at three proglacial gauging sites in NW Greenland."
**L66:** Define the acronym AWS here, and use it across the paper e.g., L90, L109 "AWS".
**L73:** As the acronym GrIS was defined earlier, you can now use it across the manuscript.
**L95:** It would be good to introduce the acronym PT for "pressure transducer" somewhere in the text.

**L113:** "pressure transducer that measures …"
**L100:** Remove the dot after "riverbank)". **L116:** Add a dot after "1.5%". **L187:** Add a dot after "2023)"
**L145:** "is achieved via"
**L184:** "IR obtained from the integrated"
**L195-196:** Maybe: "due to turbulent flow as confirmed by the time-lapse camera imagery."
**L213:** "used to calculate"
**L258:** "meteorological station"
**L259:** "difference between shortwave/solar downward and upward radiation."
**L268:** "two bank-mounted laser …"
**L315:** Remove "producing".
**L316:** "low quality" instead of "suspect quality"?
**L336:** "yields the most complete"
**L339:** Remove "timescale" after 15-min.
**L374:** Replace "Having said that," by "However,"
**L409:** "predict surface mass loss from runoff in an understudied".
**L412:** "upward/downward shortwave/solar radiation"

**Figures and Tables**
**L255:** Add a reference to Table 1 after "hybrid product)".
**L285:** Refer to Fig. 6 after "beginning in 2021".
**L295:** Refer to Fig. 7 after "June,7 2021".
**L316:** Refer to Fig. A1 after 75%.
**Figure 1:** Explain what the yellow star represents in the caption.
**Figure 2:** L438: "bedrock-dominated … to study surface runoff without …"
**Figure 4:** Spell out what PT means.
**Figure 5:** The authors could highlight when the camera data are low quality with e.g. colored shades. Please, also explain what Lidar M1 and M2 refer to in the caption. The same holds for Figs. 6 and 7 with Lidar N1 and F1.
**Table 1:** The caption should briefly describe the information listed in the Table, not provide an analysis of its content. Could you reformulate? Also explicitly state that these statistics only refer to the Minturn River location.
**Figure 8:** Use the same x-axis in all subfigures (see Net shortwave/solar radiation).
L484: "net shortwave/solar radiation".
**Appendix A:** For clarity, "by -6 m or -9 m" instead of "by one of the two values".
**L500:** "were observed in the laser rangefinder …"

**References**

Mouginot et al. (2019): https://www.pnas.org/doi/10.1073/pnas.1904242116
King et al. (2020): https://www.nature.com/articles/s43247-020-0001-2
Trusel et al. (2018): https://www.nature.com/articles/s41586-018-0752-4
Noël et al. (2020): https://agupubs.onlinelibrary.wiley.com/doi/10.1029/2020GL090471
Hofer et al. (2020): https://www.nature.com/articles/s41467-020-20011-8
Mankoff et al. (2020): https://essd.copernicus.org/articles/12/2811/2020/

---

## Author Comment (AC1)

Dr. Ciro Apollonio
Associate Editor
Geoscientific Instrumentation, Methods and Data Systems

25 July 2023

Re: Revisions to gi-2023-3

Dear Dr. Apollonio,

We are pleased to submit for your consideration our revised manuscript "New Proglacial Meteorology and River Stage Observations from Inglefield Land and Thule, NW Greenland". A stepwise, detailed response to all comments appears below.

Thank you for considering this revised manuscript for publication in *Geoscientific Instrumentation, Methods and Data Systems*. If we may provide any further explanation of our revisions or findings, please do not hesitate to contact the lead author at sarah_esenther@brown.edu.

Respectfully submitted,

Sarah Esenther
PhD Candidate
Department of Earth, Environmental & Planetary Sciences
Brown University

**RESPONSE TO REVIEWERS' COMMENTS**

**Response to Reviewer #1:**
The authors present new field instruments to measure river stage (water level) at three locations in northwest Greenland over 2019-2021. Instruments include a CF Bubbler, laser rangefinder units, pressure transducers and time-lapse cameras, recording stage values at different temporal resolution (15-min, hourly, 3-hourly) with various uncertainties and data gaps. To address this, the authors introduce a hybrid product that merges the most accurate sensor estimates from the extensively monitored Minturn River location. The authors highlight the added value of the hybrid product which yields the highest accuracy, while having the shortest data gaps among all other products. Using remote sensing (precipitation and albedo) and meteorological data (shortwave radiations, air temperature) from an automatic weather station installed at Minturn River, the authors identify the drivers of river stage changes.

I enjoyed reading the paper, which is very well written, and thoroughly describes the instruments used to estimate river stage, including uncertainties, pros and cons of each technique. The presented data set will undoubtedly be of high interest to the community. However, I missed a more elaborate analysis on the potential drivers of river stage variability in section 3.4. I also have some concerns about the "suggested purpose" of this new data set, i.e., whether it can be used for climate model evaluation. Below, the authors will find my General, Point and Stylistic comments. Based on these, I recommend **major revisions** before acceptance.

**General comments**

1. The abstract, introduction and conclusions highlight the need for new surface runoff estimates to evaluate (SMB) climate models. However, the actual river stage as measured in this study is not simulated by climate models, which estimate runoff fluxes, preventing a direct comparison. If possible, the authors could attempt to estimate runoff fluxes combining the hybrid stage product with the riverbed level and section. The resulting runoff flux data set could then be directly used for climate model evaluation. If this cannot be achieved with available measurements, I recommend reformulating the statements about model evaluation (e.g., L17-19, L39-41, L45-46, L50-51, L71-73, L408-409), and explicitly state that river stage is measured, but not the runoff flux, to avoid confusions. The manuscript title could be reformulated as: "River stage observations from …"

We agree with the reviewer that runoff flux, not runoff stage, is necessary for comparison with SMB climate models. We now clarify that river stage, not flux, is measured throughout the paper, including in the abstract (L19-21):
*"To obtain hydrological and meteorological datasets suitable for both runoff stage characterization and, pending establishment of stage-discharge curves, SMB model evaluation,"*

We also now make the application of our data clear in the the Introduction by adding the following new paragraph (L67-73):
*Analysis of runoff stage data enables evaluation of diurnal, seasonal, and annual runoff patterns as well as assessment of the relationship between these patterns and meteorological drivers. However, as SMB climate models estimate runoff flux, river stage measurements alone cannot be directly compared with SMB outputs. River stage measurements must be combined with a stage-discharge curve (established with in situ discharge measurements) and careful watershed delineation to allow for comparison between in situ runoff flux and SMB climate model runoff flux. Such discharge measurements were recently collected by the author team and are currently undergoing quality control.  These data will be presented with a remotely-sensed ice watershed delineation and SMB model outputs in a future publication.*

In the conclusion of the paper, we clarified the noted line as follows (L440-441):
*Following release of the stage-discharge curves established for the Minturn and North Rivers, these data can be used to test SMB models used to predict ocean-going ice mass loss from an understudied, rapidly changing area.*

We also revised the title to specify that river stage observations were collected, as requested ("New Proglacial Meteorology and River Stage Observations from Inglefield Land and Thule, NW Greenland").

2.  The main results section 3.4 is relatively short. The correlation analysis between the hybrid product and meteorological variables to identify drivers of river stage variability is interesting. However, no figures support the analysis, making it hard for the reader to interpret the results. The authors could consider adding scatterplots between each tested meteorological variables (xaxis) and the hybrid stage product (y-axis) and provide corresponding $R^2$ and p-values within each graph. The authors could also elaborate on the impact of e.g., precipitation (rain), air temperature, cloudiness and shortwave radiation, ice albedo on the recorded stage variability at Minturn River. This is initiated in L345-353 and L363-366, but a broader analysis would be beneficial.

We have now added scatterplots showing the observed relationships between meteorological variables and proglacial river stage as requested (figure A2). The discussion of the statistical analyses has been substantially expanded, including discussion of the impact of precipitation, air temperature, cloudiness and shortwave radiation, and ice albedo on stage, as follows (L373-417):
*After applying a conservative Bonferroni correction, air temperature (at all lag steps), upward/downward/net solar radiation, ice sheet/proglacial/full watershed albedo, and ice sheet/full watershed snow cover were significantly correlated with hybrid stage (Fig. A2). As air temperature at all lag steps (0-2 day) showed similar, strong correlation and significance values, 2-day average (lag 0-1 day) and 3-day average (lag 0-2 day) air temperature variables were introduced. The strongest predictor from each group of correlated variables was selected. This yielded 2 day average air temperature (p-value=$1.17 \times 10^{-62}$, $R^2$=0.650) and ablation zone albedo (p-value=$4.10 \times 10^{-71}$, $R^2$=0.620) as the independent predictors included in the multivariate linear*

[revised manuscript text omitted]

3. The discussion section currently repeats some information presented in the results section (e.g., L368-386), suggesting that these sections could be merged.

We removed L368-386 and merged the discussion section with the results section as requested. The previous final paragraph of the discussion section, summarizing the findings of the stage/meteorological variable assessment, was appended to 3.4 Watershed Characterization and the limitations paragraph was given its own section (3.5 Limitations) to streamline this part of the paper.

**Point comments**

**L36:** Runoff and solid ice discharge are the two major contributors to GrIS mass loss, I suggest reformulating as: "Besides solid ice discharge, climate change-induced meltwater runoff is a dominant driver of Greenland ice sheet (GrIS) mass loss that is projected to increase …" or something equivalent. The authors could consider providing more recent references including e.g., Mouginot et al. (2019), King et al. (2020) on the contributors to GrIS mass loss, and Trusel et al. (2018), Noël et al. (2020) or Hofer et al. (2020) on the meltwater runoff increase in the 21st century.

Thank you for pointing out this important clarification. We adopted the suggested sentence and the suggested references (L36-39).

**L38-39:** What do you mean by mass budget residual? In climate models, runoff is estimated using tipping bucket snow models that represent surface melt (solving the surface energy budget (SEB)), meltwater retention and refreezing in firn layers, and subsequent runoff. Could you clarify the statement?

We clarified these statement as requested (L38-41):
*However, current climate models typically calculate runoff as a residual term in surface mass balance budgets (van Dalum et al., 2021). As runoff represents rain and meltwater that is not refrozen or retained in the firn, errors in the surface energy balance terms used to calculate*

*melt/refreezing or in any of the other surface mass balance terms propagate to error in the subsequent runoff term calculation.*

**L39-41:** I am confused here, as the river stage measurements cannot be directly compared to modeled runoff fluxes from climate models. See also General comment #1.

We added a sentence in the abstract (L19-21) and a paragraph in the introduction (L67-73) clarifying that the runoff stage measurements require forthcoming stage-discharge rating curve estimates. More detail is provided in the General comment #1 response

**L45-47:** Mankoff et al. (2020) compared modeled and measured runoff from different locations, including Qaanaaq in NW Greenland and Zackenberg in NE Greenland. Please, clarify.

As requested, we clarified that the proglacial rivers have been gauged outside of SW Greenland, including by Mankoff et al. (2020). L47-50:
*Despite this need for in situ hydrological measurements, only a small handful of GrIS proglacial rivers have been gauged outside of SW Greenland (Ploeg et al., 2021; Mankoff et al., 2020; Mernild et al., 2008) and the majority of modelled runoff evaluation studies have been conducted in SW Greenland (Smith et al., 2017; Mernild et al., 2011; Mernild et al., 2018; Cooper et al., 2018; Smith et al., 2015).*

**L61-62:** Runoff discharge measurements are available from Qaanaaq in NW Greenland in Mankoff et al. (2020). Please, clarify.

We clarified the sentence as requested (L64-65):
*To our knowledge no in situ hydrological observations of this length (3+ years) are currently available for NW Greenland, an exceedingly cold and remote region.*

**L123:** I am not sure about the journal policy on unpublished references. Either remove or use "personal communication". The same holds across the manuscript.

This reference has been published since the manuscript was submitted. The references have been updated throughout the body and in the citations as requested (L62, 135, 241, 242, 245, 267, 288, 294, 335, 577).

**L182-183:** Do you mean that the two other watersheds are too small to derive significant statistics? Please, clarify.

We clarified that the watersheds were omitted because the significance of the relationships contained more uncertainty for these watersheds, given the discrepancy between the watershed sizes and the resolutions of the remote sensing products as follows (L194-196):
*Due to the small size of the North River and Fox Canyon River watersheds relative to the spatial resolutions of the remote sensing datasets, these datasets are less representative of conditions*

*within the watersheds and statistical analyses over these watersheds are omitted from this paper.*

**L209-210:** For clarity, the authors could state that laser rangefinder M1, M2, N1 and F1 located in the three regions will be referred to as Lidar M1, M2, N1 and F1 in Figures 5-7. This should be done after introducing each of these sensors for the first time.

We revised the introduction of the laser rangefinders to note their names, as requested (L111, 155, 176):
*River stage measurements are complemented by two custom-built, bank-mounted, oblique-looking Laser rangefinder systems, M1 and M2 (one on each riverbank). (Fig. 3).*

*On the right bank of the North River, a third custom-built, bank-mounted LT Trusense S200 laser rangefinder ranging system, N1,*

*At the Fox Canyon River, a fourth custom-built, bank-mounted LT Trusense S200 laser rangefinder ranging system, F1,*

We also added a note that the regions will be referred to as Lidar M1, M2, N1, and F in Figures 5-7 when the first figure is cited, as requested (L251):
*Laser rangefinders M1, M2 (Minturn River), N1 (North River), and F1 (Fox Canyon River) will be referred to as Lidar M1, M2, N1, and F1, respectively, in Figures 5-7.*

**L216-219:** Could you refer to the terms of equation 1 in the main text: e.g. "surface below the laser box ($Z_{Lidar Box}$) … distance to water surface (Median Lidar Distance) … vertical angle of the laser range finder ($\theta_{Lidar Box}$)"

Completed as requested (L227):
*Simple trigonometry was used to compute the vertical distance of the water surface below the laser box ($Z_{Lidar Box}$) using the measured distance to the water surface (Median Lidar Distance) and the vertical angle of the laser range finder ($\theta_{Lidar Box}$).*

**L235:** Could you highlight these lowest 75% stage values in Fig. 5 e.g., using colored shades? The reader can thus directly notice when the camera data set has the largest uncertainty.

To maintain clarity with the other colors of stage records in the plot, we kept the camera record all the same color and instead denoted the reduced quality measurements with X markers.

**L246:** What do you mean by "predict CF Bubbler"? Fill the gap in that data set?

Clarified with the suggested language (L258-259):
*fitting the Level TROLL PT record with a linear model to fill gaps in the CF Bubbler data*

**L255-265:** The assessment described here is not supported by any figure/table and is only briefly discussed in section 3.4. See also General comment #2.

We added the requested scatterplots between stage and the meteorological variables with $R^2$ and p-values, citing the new (L270-271):
*Correlation coefficients, p-values, and scatterplots were also produced for each variable available from the meteorological station and remote sensing data sets (Fig. A2).*

Discussion of the ANOVA assessment was described in A2 (L279-284):
*ANOVA tests were performed on the air temperature, ice sheet albedo, precipitation, net radiation, downward radiation, and hybrid stage to assess differences in means between early melt season (~day of year 190 to 205) for each year (2019, 2020, 2021). The early date selection was based on visual inspection of the runoff stage records; qualitatively, 2019 and 2020 appear to follow similar patterns of early high runoff stage and appearance of a diurnal signal, while runoff was lower and emergence of diurnal signal was delayed in 2021. To investigate these differences, we limited our early season assessment to the period between day 289 (the first day records are available in all years) and day 205 (the latest onset of a pronounced diurnal signal).*

Discussion of the findings was discussed more thoroughly in section 3.4 (full changed text included in General comment #2).

**L289:** Do you mean "July 15" instead of "June 15"?

Thank you for pointing out this error. Changed as requested.

**L343:** In section 3.4, could you refer to Figures 5-7 where appropriate, and explicitly state the range of discussed "days of the year".

Changed as requested. The days of the year are stated explicitly in L386-387:
*day 189 (July 7/8, the first day a stage record is available in all years) to day 205 (July 23/24, the latest onset of a pronounced diurnal signal)*

Figures 5-7 are now referenced in section 3.4 in lines 369, 370, and 371.

**L345:** The authors should state that the AWS data are only available at Minturn River.

Changed as requested (L364):
*Weather conditions at the Minturn River AWS were similar between years*

**L348:** "diurnal cycle of river stage"?

Changed as requested.

**Style**

**L20:** "SMB model evaluation" instead of "validation". L40: "evaluating", L46: "evaluation" and L51:
"Evaluate".

Changed as requested.

**L32-33:** For conciseness: "hydrological observations that are freely available …"

Changed as requested.

**L45:** What do you mean by "SMB runoff", "modeled runoff"? Please, clarify.

Changed as requested.

**L50:** "quantify runoff drainage to the ocean and evaluate SMB models."

Changed as requested.

**L60:** "thus require"

Changed as requested.

**L64:** Maybe: "This paper describes new hydrometric sensor installations, and the resulting 3-year time series (2019-2021) of river stage (water level) at three proglacial gauging sites in NW Greenland."

Changed as requested.

**L66:** Define the acronym AWS here, and use it across the paper e.g., L90, L109 "AWS".

Changed as requested.

**L73:** As the acronym GrIS was defined earlier, you can now use it across the manuscript.

Changed as requested.

**L95:** It would be good to introduce the acronym PT for "pressure transducer" somewhere in the text.

Changed as requested, PT defined the at the first occurrence of "pressure transducer" (L59)

**L113:** "pressure transducer that measures …"

Changed as requested.

**L100:** Remove the dot after "riverbank)". **L116:** Add a dot after "1.5%". **L187:** Add a dot after "2023)" **L145:** "is achieved via"

Changed as requested.

**L184:** "IR obtained from the integrated"

Changed as requested.

**L195-196:** Maybe: "due to turbulent flow as confirmed by the time-lapse camera imagery."

Changed as requested.

**L213:** "used to calculate"

Changed as requested.

**L258:** "meteorological station"

Changed as requested.

**L259:** "difference between shortwave/solar downward and upward radiation."

Changed as requested.

**L268:** "two bank-mounted laser …"

Changed as requested.

**L315:** Remove "producing".

Changed as requested.

**L316:** "low quality" instead of "suspect quality"?

Changed as requested.

**L336:** "yields the most complete"

Changed as requested.

**L339:** Remove "timescale" after 15-min.

Changed as requested.

**L374:** Replace "Having said that," by "However,"

Paragraph removed in alignment with General Comment #3.

**L409:** "predict surface mass loss from runoff in an understudied".

Changed as requested.

**L412:** "upward/downward shortwave/solar radiation"

Changed as requested.

**Figures and Tables**

**L255:** Add a reference to Table 1 after "hybrid product)".

Changed as requested.

**L285:** Refer to Fig. 6 after "beginning in 2021".

Changed as requested.

**L295:** Refer to Fig. 7 after "June,7 2021".

Changed as requested.

**L316:** Refer to Fig. A1 after 75%.

Added a reference to Figure 5a after 75%.

**Figure 1:** Explain what the yellow star represents in the caption.
Changed as requested:

*Thule Air Force Base, which collects airport meteorological records, is indicated by the star.*

**Figure 2:** L438: "bedrock-dominated … to study surface runoff without …"

Changed as requested.

**Figure 4:** Spell out what PT means.

Changed as requested.

**Figure 5:** The authors could highlight when the camera data are low quality with e.g. colored shades. Please, also explain what Lidar M1 and M2 refer to in the caption. The same holds for Figs. 6 and 7 with Lidar N1 and F1.

To improve clarity with the other colored records, the camera stage record was kept the same color throughout but have now denoted the reduced quality measurements, as requested, with an X marker. Captions were changed as requested:
*Figure 5. Hydrographs of stage at the Minturn River (a) were produced by two bank-mounted laser rangefinder devices (Lidar M1 and Lidar M2),*

*Figure 6. Records of stage at the North River are available from a bank-mounted laser rangefinder unit (Lidar N1),*

*Figure 7. Stage records from 2019 to 2021 at the Fox Canyon River are available from a bank mounted laser rangefinder unit (Lidar F1).*

**Table 1:** The caption should briefly describe the information listed in the Table, not provide an analysis of its content. Could you reformulate? Also explicitly state that these statistics only refer to the Minturn River location.

Changed as requested:
*Table 1. Coefficients of determination ($R^2$) and root-mean-square-errors (RMSE) of different river stage sensors relative to the CF Bubbler vented pressure transducer record (considered the gold standard for stage accuracy, $\pm$0.003m).[20] Sensor pros/cons, including data completeness at 15-min, hourly, and daily timesteps, are summarized.*

**Figure 8:** Use the same x-axis in all subfigures (see Net shortwave/solar radiation).

Changed as requested

L484: "net shortwave/solar radiation".

Caption changed as requested.

**Appendix A:** For clarity, "by -6 m or -9 m" instead of "by one of the two values".

Caption changed as requested.

**L500**: "were observed in the laser rangefinder …"

Caption changed as requested.

**Response to Reviewer #2:**
The paper by Esenther et al. describes newly installed hydrometric sensors in northwest Greenland and the results and performance of the first three years of operation (2019 to 2021). The installed devices aim to better monitor the discharge of the Greenland Ice Sheet. The sensor systems include standardized devices and novel methods, such as laser rangefinder stage measurement for remote river gauges. The devices, data acquisition, processing and data analysis are described in detail, making it understandable for anyone who wants to set up a similar system or use the data.

We thank the reviewer for their supportive comments.

For me, this is an excellent read, well-structured and valuable contribution to the community. The instruments' description, interaction, and benefits are very detailed without making the text excessively long. Moreover, I see the first analysis of the data as robust, which speaks to the positive performance of the sensors. The only thing that could be explained a little better is to present how exactly the specific runoff quantities can be included in the calculation of the SMB, which ultimately serves as model input. Otherwise, the article as it stands is a great addition to the journal Geoscientific Instrumentation, Methods and Data Systems.

This comment was brought up by the other reviewer and we agree that flux, rather than stage alone, is necessary for comparison with RCM/SMB runoff. We have now clarified that river stage, not flux, is measured throughout the abstract and paper, with details given in General comment #1 of our response to reviewer #1.